

# 1 Estimating ocean heat content from the ocean thermal expansion
# 2 parameters using satellite data

Vijay Prakash Kondeti[1], Palanisamy Shanmugam[1]
[1]Ocean Optics and Imaging Laboratory, Department of Ocean Engineering, Indian Institute of Technology Madras, Chennai-
600036, India
*Correspondence to*: Palanisamy Shanmugam (pshanmugam@iitm.ac.in)
**Abstract.** Ocean heat content (OHC) is a depth-integrated physical oceanographic variable used to precisely measure ocean
warming. Because of the limitations associated with in-situ CTD data and Ocean Reanalysis system products, satellite-based
approaches have gained importance in estimating the daily to decadal variability of OHC over the vast oceanic region. Efforts
to minimize the biases in satellite-based OHC estimates are needed to realize the actual response of the ocean to the brunt of
climate change. In the current study, an attempt has been made to better implement the satellite-based ocean thermal expansion
method to estimate OHC at 17 depth extents ranging from the surface to 700m. To achieve this objective, an artificial neural
network (ANN) model was developed to derive thermosteric sea level (TSL) from a given dataset of sea surface temperature,
sea surface salinity, geographical coordinates, and climatological TSL. The model-derived TSL data were used to estimate
OHC changes based on the thermal expansion efficiency of heat. Statistical analysis showed high correlation coefficients and
low errors in satellite-derived TSL / OHC at 700 m water depth (N 388469, R 0.9926 / 0.9922, RMSE 1.16 m / 1.56 GJ m$^{-2}$,
MBE -0.1917 m / -0.2400 GJ m$^{-2}$, MBPE -0.4560% / -0.0290%, MAE 0.763 m / 1.029 GJ m$^{-2}$, and MAPE 2.34% / 0.13%)
and nearly similar results at the remaining depth extents. These results suggest that the proposed ANN models are capable of
accurately estimating OHC changes on real-time data and three-dimensional distribution patterns of depth-integrated OHC
trends in the global ocean. In addition, the first-ever attempt to estimate the ocean thermal expansion component (*i.e.*, TSL)
from satellite data was successful and the model-derived TSL can be used to obtain high-end sea-level rise products in the
global ocean.

## 23 1. Introduction

Owing to the vast heat capacity and spatial coverage, the oceans balance the planet's temperatures by absorbing 93% of the
excess atmospheric heat caused by the greenhouse effect and global warming (Abraham et al., 2013; IPCC, 2014; Roemmich
et al., 2015; Riser et al., 2016; Trenberth et al., 2016; Meyssignac et al., 2019). A precise understanding of the depth-wise
penetration of this heat and its accumulation in the upper oceanic layers is inevitable (Liang et al., 2015; Baxter, 2016; IPCC,
2022). Ocean heat content (OHC), a depth-integrated physical oceanographic variable that refers to the amount of heat energy
accumulated between any two depths, has gained attention in various studies of the Earth Energy Imbalance (Von Schuckmann





et al., 2016; Trenberth et al., 2016; Cheng et al., 2017; Meyssignac et al., 2019; Cheng et al., 2022). Thus, accurate estimation
of OHC changes at various depth extents is vital and the motivation of the current study.

To obtain a complete picture of OHC changes at different depths, the direct measurements of in-situ conductivity,

temperature, and depth (CTD) profiles are necessary. These in situ measurements of the ocean properties are limited in terms
of depth and spatial coverages, leading to the biased global reconstruction of OHC estimates owing to the sparse measurement
data and spatial coverage deficiencies (Jagadeesh et al., 2015; Meyssignac et al., 2019; Marti et al., 2022). However, the in-
situ CTD profile measurements have been used to develop and validate the different OHC models (Momin et al., 2011;
Jagadeesh et al., 2015; Su et al., 2020; Prakash and Shanmugam, 2022). In addition, synthetic CTD profile data generated by
the Ocean Reanalysis systems (ORA) have been used to compute OHC variability in spatial and temporal scales (Balmaseda
et al., 2015; Palmer et al., 2017). More recently, satellite-based methods have become crucial to overcome the limitations in
the in-situ measurements of OHC changes, to ensure the OHC trend at a global scale, and to understanding the evolution of
the Earth's climate system (Meyssignac et al., 2019; Prakash and Shanmugam, 2022).

The existing satellite-based OHC algorithms can be broadly grouped into three approaches based on the employed

principles/parametrizations: (i) internal tide oceanic tomography (ITOT), (ii) ocean net surface heat fluxes, and (iii) ocean
thermal expansion. Apart from these approaches, research is exploring ways to make use of tidal magnetic satellite observations
(Irrgang et al., 2019), electrical conductance (Trossman and Tyler, 2019), and atmospheric oxygen & carbon dioxide
concentrations (Resplandy et al., 2018) to infer OHC changes. The ITOT technique involves correlating the satellite altimeter-
derived internal tide phase changes with ocean warming to estimate the OHC variability. This technique is still at the proof-
of-concept level and the associated challenges remain to be addressed (Zhao, 2016; Meyssignac et al., 2019). The OHC
estimation through the ocean net surface heat fluxes employs several assumptions and approximations in deriving the input
parameters to compute the radiative and turbulent heat fluxes, which in turn leads to a higher uncertainty of global OHC
changes (Wild et al., 2015; L'Ecuyer et al., 2015; Meyssignac et al., 2019). On the other hand, the ocean thermal expansion
method is a promising technique for the estimation of OHC by considering the thermosteric sea level (TSL) and expansion
efficiency of heat (EEH). Numerous satellite-based OHC models have been developed based on the sea surface height anomaly
data from altimeters, water mass change equivalent sea level anomaly data from the Gravity Recovery and Climate Experiment
mission (GRACE), sea surface temperature from the various radiometers onboard satellites, and wind speed/stress from
scatterometers/numerical weather models. Pioneering work done by White and Tai (1995), Chambers et al. (1997), Polito et
al. (2000), and Sato et al. (2000) have attempted to implement the ocean thermal expansion method based on a relationship
between OHC and satellite altimeter-based sea surface height anomaly (SSHA). It should be mentioned that regardless of the
source, the density of seawater changes when it is subjected to heating/cooling, and it eventually reflects in sea surface
topography. The SSHA data recorded by the satellite altimeters comprise the sea surface topography changes due to tides,
atmospheric pressure, salinity (haline), and barotropic flows along with the thermal effects. The SSHA changes due to the tides
and atmospheric pressure can be corrected, but the effects of salinity and barotropic flows remain unresolved with the OHC
estimates produced by Wang and Tai (1995) and Chambers et al. (1997). Sato et al. (2000) have introduced a haline correction





factor as the integral product of the haline contraction coefficient and salinity anomaly from in-situ CTD profile data. Owing
to the limitations associated with in-situ data, the in-situ-based haline correction cannot be applied to satellite altimeter-based
SSHA data while correlating with the space and time-varying OHC data. Jayne et al. (2003) have proposed the Alt-GRACE
approach to resolve the effect of barotropic flows in sea surface topography by subtracting the satellite gravimetry-derived
water mass change component from SSHA data. Though the Alt-GRACE approach has improved the accuracy of satellite-
based OHC estimates compared to Wang and Tai (1995), Chambers et al. (1997), Polito et al. (2000), and Sato et al. (2000),
the issues associated with the haline effects and other approximations on the ocean thermal expansion coefficient and seawater
density data have led to significant uncertainties in satellite-based OHC estimates. With the advancement of artificial
intelligence, several researchers have attempted to model OHC by directly relating it with the satellite-based parameters by
using deep-learning regression techniques (Jagadeesh and Ali, 2006; Momin et al., 2011; Chacko et al., 2015; Jagadeesh et al.,
2015; Su et al., 2020, 2021; Marti et al., 2022). These deep-learning models have oversimplified the OHC problem by
neglecting the effects of salinity and barotropic flows. In addition, no previous work have accounted for the space and time-
varying nature of the ocean thermal expansion coefficient and seawater density in OHC computations. The other common
drawbacks with the existing work are discussed in Sect. 4.3. Consequently, there is a need for developing a satellite-based
model to accurately implement the ocean thermal expansion method to estimate OHC by resolving all the issues associated
with salinity variation, barotropic flows, ocean thermal expansion, seawater density, choice of temperature and its units.

Given the above background, we have made a major attempt to develop and implement a satellite-based ocean thermal

expansion model for estimating OHC changes at various depth extents such as 20 m, 30 m, 40 m, 50 m, 100 m, 150 m, 200 m,
250 m, 300 m, 350 m, 400 m, 450 m, 500 m, 550 m, 600 m, 650 m, and 700 m. For this, artificial neural network (ANN)
architectures were developed to estimate TSL for the given sea surface temperature (SST), sea surface salinity (SSS),
geographical coordinates, and climatological TSL. The model-derived TSL estimates were then used to estimate OHC changes
by accounting the expansion efficiency of heat. The proposed models are capable of estimating TSL and OHC accurately at
multiple depth extents. The robustness of the new models was tested by comparison of satellite-derived TSL and OHC with
in-situ data.
**2. Data**
For this study, in-situ CTD profile data (collected by Argo floats) were obtained from the World Ocean Database-2018 of the
NOAA's National Centers for Environmental Information Data Archive for the period of 2005-2020 (Boyer et al., 2018a).
These data have been extensively used by the research community for various ocean applications (Levitus et al., 2009; Momin
et al., 2011; Levitus et al., 2012; Cheng et al., 2014; Roemmich et al., 2015; Jagadeesh et al., 2015; Su et al., 2020). The World
Ocean Database (WOD) comprises the oceanographic data of diverse biogeochemical parameters that have been collected by
various institutions, agencies, individual researchers, and data recovery initiatives. The quality-controlled CTD profile data
(*accepted_value* flag) of standard depth levels recommended by the International Association of Physical Oceanography



(1936) were considered in this study to compute the $TSL_d$ and $OHC_d$ parameters and to obtain the SST and SSS data. The
standard depth levels considered for deriving the TSL and OHC are given as 20 m, 30 m, 40 m, 50 m, 100 m, 150 m, 200 m,
250 m, 300 m, 350 m, 400 m, 450 m, 500 m, 550 m, 600 m, 650 m, and 700 m. The in-situ $TSL_d$ and $OHC_d$ parameters were
computed by applying the integration formula (Eqs. 1 & 2) on the CTD profile data of depth range from the ocean surface to
the respective standard depth (d) as well as the SST and SSS data corresponding to the ocean surface. Similarly, the
climatological parameters such as $TSL_{clim,d}$ and $OHC_{clim,d}$ were computed from the monthly climatological temperature and
salinity data of 41 vertical levels obtained from the World Ocean Atlas-2018 (WOA) (Boyer et al., 2018b). The theoretical
considerations of computing OHC change at a depth can be found in Prakash and Shanmugam (2022) (Prakash and
Shanmugam, 2022), which were adopted in this study. The Gibbs-SeaWater (GSW) Oceanographic Toolbox of TEOS-10
(IOC et al., 2010) was used to compute the in-situ-based parameters including
$OHC_d = \int_0^d \rho C_P \Theta \, dz$                      (1)
$TSL_d = \int_0^d \alpha \Theta \, dz$                       (2)
where $OHC_d$ refers to the heat energy accumulated in an oceanic layer of depth range from the surface to a stipulated depth
(d) and is given in the units of joules per unit area (J m$^{-2}$). Similarly, $TSL_d$ (in meters) refers to the thermosteric sea level
integrated from the surface to a stipulated depth (d). And, $\Theta$ is the conservative temperature in K (derived from in-situ
temperature, absolute salinity, and pressure), $\rho$ is the seawater density in kg m$^{-3}$ (derived from the conservative temperature,
absolute salinity, and pressure), $C_P$ is the specific heat capacity (= 3991.87 J kg$^{-1}$ K$^{-1}$), and $\alpha$ is the thermal expansion coefficient
in K$^{-1}$ (derived from the conservative temperature, absolute salinity, and pressure).
Python programming was used to prepare the individual databases for all the standard depth levels by extracting CTD
profile data from the WOD and WOA NetCDF files with the help of NetCDF4, NumPy, Pandas, and GSW libraries. Each
database was divided into two datasets, one for the model development spanning from 2005-2016 and one for validating the
model spanning from 2017-2020, by ensuring a well distribution in spatiotemporal scales over the global open ocean. The
spatial distribution of data points used to model $TSL_{700}$ and $OHC_{700}$ is shown in Fig. A1. The in-situ CTD profiles of depth
coverage shallower than 700 m are also included in this process of deriving the TSL and OHC of other depth extents. Indeed,
the number of CTD profiles and their distribution in global oceans is higher than the CTD profile density as shown in Fig. A1.
**3. Methodology**
**3.1. Theoretical formulations**
Ocean thermal expansion is the best proxy to model the heat content accumulated in an oceanic layer. Unlike freshwater,
seawater expands when it warms and contracts when it cools for temperatures above its freezing point. The volumetric
expansion of seawater is non-isotropic in nature due to the differences in the degree of constraint in different directions. In a
vertical direction, atmospheric pressure exerts a normal force on the seawater parcel at the surface. The magnitude of this





normal/vertical force is less compared to the horizontal forces exerted by physical barriers such as continental boundaries and
geographic features on the ocean floor. It allows the ocean thermal expansion of seawater in the vertical direction rather than
the horizontal direction, as the seawater is less constrained in the vertical direction compared to the horizontal direction. The
amount of change in seawater volume in response to the net warming/cooling depends on the absolute conservative temperature
and ocean thermal expansion coefficient (Eq. 2). Following are the GSW functions (Eqs. 3-5) (IOC et al., 2010) involved in
the calculation of TSL (Eq. 2) for the given set of measured temperature (T), practical salinity (SP), pressure (P), longitude
(x), and latitude (y).
$Absolute\ salinity\ (SA) = gsw.SA\_from\_SP\ (SP, P, x, y)$        (3)
$\Theta = gsw.CT\_from\_T(SA, T, P)$        (4)
$\alpha = gsw.Alpha(SA, \Theta, P)$        (5)
Hence, an attempt has been made in this study to model TSL as a function of SST, SSS, and geographical coordinates. The
existing correlations between the proposed input parameters and the targeted output parameter were explored by employing
in-situ-based data used in the model development process (Fig. 1).

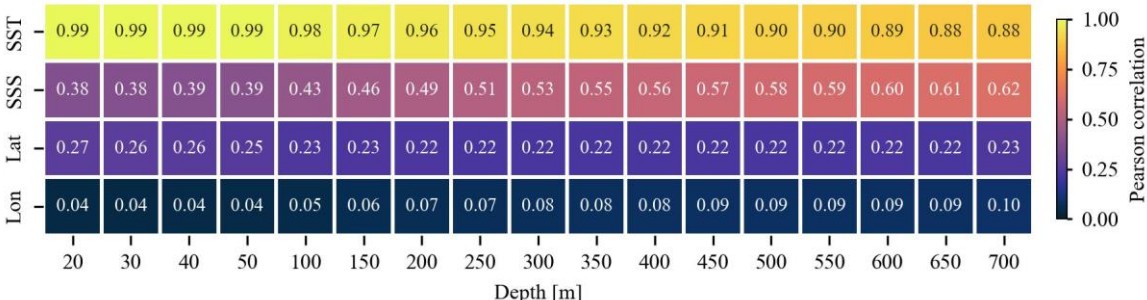


**Figure 1.** Heatmap showing the Pearson correlation coefficients between the input parameters (*i.e.*, SST, SSS, and
geographical coordinates) and the output parameter (TSL) of various depth extents.

It is observed that SST has an almost one-to-one correlation with TSL at shallower depth extents, and can be solely

used to model the thermal expansion of upper oceanic layers. Despite a decreasing trend in correlation strength when moving
towards a deeper depth, SST plays a primary role in accounting for TSL variations at deeper depths, because of its strong
correlations with TSL. Observed weaker correlations between SSS and TSL which are plausible owing to the salinity's
secondary role in TSL variations as compared to the temperature variable. However, an increasing trend in correlation
coefficients between SSS and TSL is observed towards the deeper depth extents. Hence, SST and SSS are complementary to
each other in resolving the TSL variations, and their combination plays a major role in modelling TSL of all depth extents
considered in this study. Apart from these physical parameters, absolute salinity used in the computation of seawater density,
conservative temperature, and ocean thermal expansion coefficient is a function of geographical coordinates along with
practical salinity and pressure (Eq. 3). By considering all these theoretical considerations and observed correlations, an attempt
has been made to model TSL of various depth extents by employing SST, SSS, and geographical coordinates as the input





parameters along with the climatological TSL (Fig. 2). Here, $TSL_d$ is an external manifestation of $OHC_d$ stored in an oceanic
layer based on $EEH_d$ (Eq. 6). The model-derived TSL is further used to estimate OHC changes (as shown in Fig. 2 along with
climatological OHC) as follows,
$$OHC_d = \frac{TSL_d}{EEH_d}$$ (6)
where *EEH* is a conversion factor that explains the relationship between the relative changes in ocean heat content and the
corresponding seawater thermal expansion. As it varies as a function of temperature, salinity, and pressure, EEH is not a
constant value over the global ocean. Hence, ANN modelling is employed in this study to derive OHC from TSL by accounting
the complex variations in EEH.

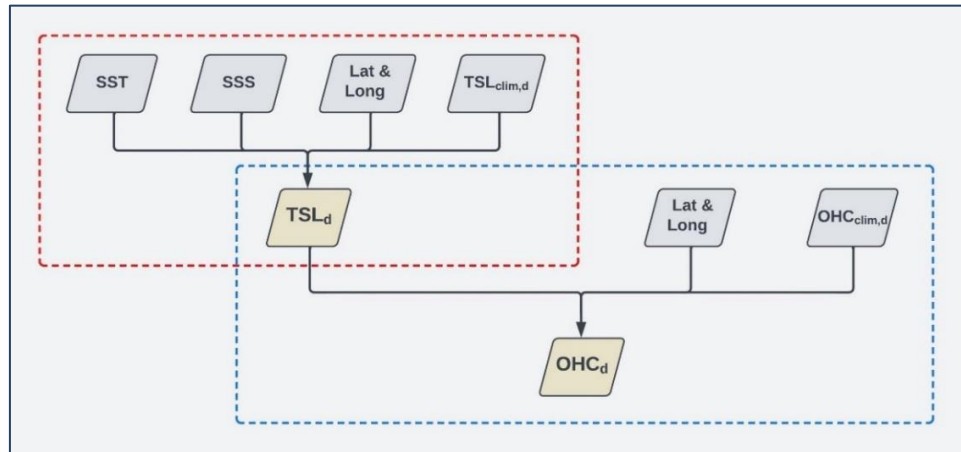

**Figure 2.** Flow chart representing the parameters involved in TSL and OHC modelling. The red and blue dashed boxes
represent the TSL and OHC frameworks employed in ANNs, respectively.

**3.2. ANN model description**
This section explains the various steps and architectures involved in the ANN modelling of TSL and OHC. The multilayer
perceptron regressor algorithm of deep neural networks was used to model both TSL and OHC (Pedregosa et al., 2011). It is
observed that the input data of geophysical parameters are given in different units and scales. The range and order of SST,
SSS, latitude, and longitude data are -1.8 °C to 34.15 °C & $O(10^1)$, 2.53 PSU to 40.45 PSU & $O(10^1)$, -76° to 80° & $O(10^1)$,
and -180° to 180° & $O(10^2)$, respectively. In addition, the range and order of $TSL_{clim,d}$ and $OHC_{clim,d}$ are also distinct and vary
with water depth. Hence, the input data were normalized using the StandardScaler class of Scikit-Learn and feed-forwarded
through the neural networks. This StandardScaler normalizes the raw data to ensure the mean and standard deviation of each
input parameter as 0 and 1, respectively. It allows the ANN model to focus on the relative importance and relationships between
the input parameters rather than their magnitude. The standardized input data were injected into the corresponding neurons in
the input layer and forward propagated through the hidden layers and then the output layer by applying the random weights
and rectified linear unit (ReLU) activation function at each neuron. The mathematical formulations and schematic



representation related to ANN architecture are shown in Fig. 3. The model outputs were compared with the actual data and
computed mean squared error (MSE) using a loss function (Eq. 7). In addition, L2 regularization ($\alpha_{L2}$) was employed to add
a penalty term to the loss value to prevent overfitting. The observed error was then backpropagated through the network to
update weights and biases using the Adam optimizer based on the learning rate and gradient of the error (see Eq. 8 in Prakash
and Shanmugam, 2022). This process is repeated until the validation score improves more than 0.0001.
$MSE = \frac{1}{N}\sum(Y_{pred,i} - Y_{act,i})^2$ (7)
where N is the number of samples, $Y_{pred,i}$ is the predicted data, and $Y_{act,i}$ is the actual data. The model development work was
carried out by employing both the input and output parameters from the in-situ sources. It enables the ANN models to
implement the input data of any remote sensing sources to produce OHC estimates subject to the reliability and accuracy of
those data sources. The particle swarm optimization technique (Kennedy and Eberhart, 1995; Shi and Eberhart, 1998) was
employed for hyperparameter tuning, and the hyperparameters' combinations corresponding to each modelling depth are
presented in Table 1. The Joblib module of Scikit-Learn library was used to save all the TSL and OHC models of various
depths considered in this study, and the same module was used to load the TSL and OHC models of desired depth with the
help of a unified Python script.




















**Table 1.** The ANN model hyperparameters employed in TSL (regular font) and OHC (bold font) modelling of various depth extents.

| Depth (m) | Hidden layers | Batch size | $\alpha_{L2}$ | Learning rate | No. of iterations |
|---|---|---|---|---|---|
| 20 | 38, 10, 55 | 178 | 0.00422 | 0.0004 | 14 |
| | **49, 12, 34** | **183** | **0.09023** | **0.0001** | **26** |
| 30 | 100, 97, 36 | 165 | 0.00001 | 0.0001 | 14 |
| | **11, 50, 55** | **58** | **0.00079** | **0.0001** | **16** |
| 40 | 64, 71, 5 | 106 | 0.00001 | 0.0001 | 16 |
| | **57, 89, 46** | **148** | **0.09691** | **0.0001** | **19** |
| 50 | 64, 99, 30 | 241 | 0.01478 | 0.0001 | 17 |
| | **56, 59, 10** | **139** | **0.07188** | **0.0001** | **22** |
| 100 | 70, 100, 100 | 256 | 0.00001 | 0.0009 | 30 |
| | **25, 36, 63** | **256** | **0.03556** | **0.0016** | **44** |
| 150 | 47, 83, 92 | 60 | 0.00001 | 0.0005 | 34 |
| | **49, 77, 28** | **69** | **0.05176** | **0.0318** | **16** |
| 200 | 100, 100, 16 | 256 | 0.00315 | 0.0022 | 33 |
| | **27, 48, 67** | **202** | **0.05638** | **0.0367** | **18** |
| 250 | 56, 82, 67 | 174 | 0.00001 | 0.0019 | 39 |
| | **2, 100, 77** | **73** | **0.00001** | **0.0037** | **22** |
| 300 | 83, 28, 74 | 128 | 0.00001 | 0.0028 | 36 |
| | **48, 92, 10** | **87** | **0.01364** | **0.0459** | **12** |
| 350 | 85, 25, 67 | 128 | 0.04606 | 0.0013 | 20 |
| | **27, 53, 48** | **141** | **0.08585** | **0.0851** | **14** |
| 400 | 89, 75, 96 | 64 | 0.04859 | 0.0007 | 26 |
| | **49, 1, 80** | **138** | **0.00001** | **0.0031** | **20** |
| 450 | 51, 83, 95 | 128 | 0.08582 | 0.0005 | 42 |
| | **47, 27, 52** | **32** | **0.00263** | **0.0055** | **24** |
| 500 | 71, 100, 62 | 128 | 0.00001 | 0.0012 | 27 |
| | **45, 100, 63** | **126** | **0.05162** | **0.0607** | **15** |
| 550 | 47, 89, 91 | 256 | 0.00843 | 0.0011 | 44 |
| | **64, 75, 78** | **114** | **0.05176** | **0.0634** | **15** |
| 600 | 98, 65, 6 | 16 | 0.00001 | 0.0001 | 48 |
| | **63, 17, 10** | **180** | **0.04654** | **0.0538** | **23** |
| 650 | 100, 69, 75 | 16 | 0.00001 | 0.0001 | 18 |
| | **53, 74, 40** | **176** | **0.07072** | **0.0048** | **20** |
| 700 | 98, 37, 37 | 164 | 0.04262 | 0.0015 | 32 |
| | **83, 63, 79** | **216** | **0.01217** | **0.0742** | **19** |






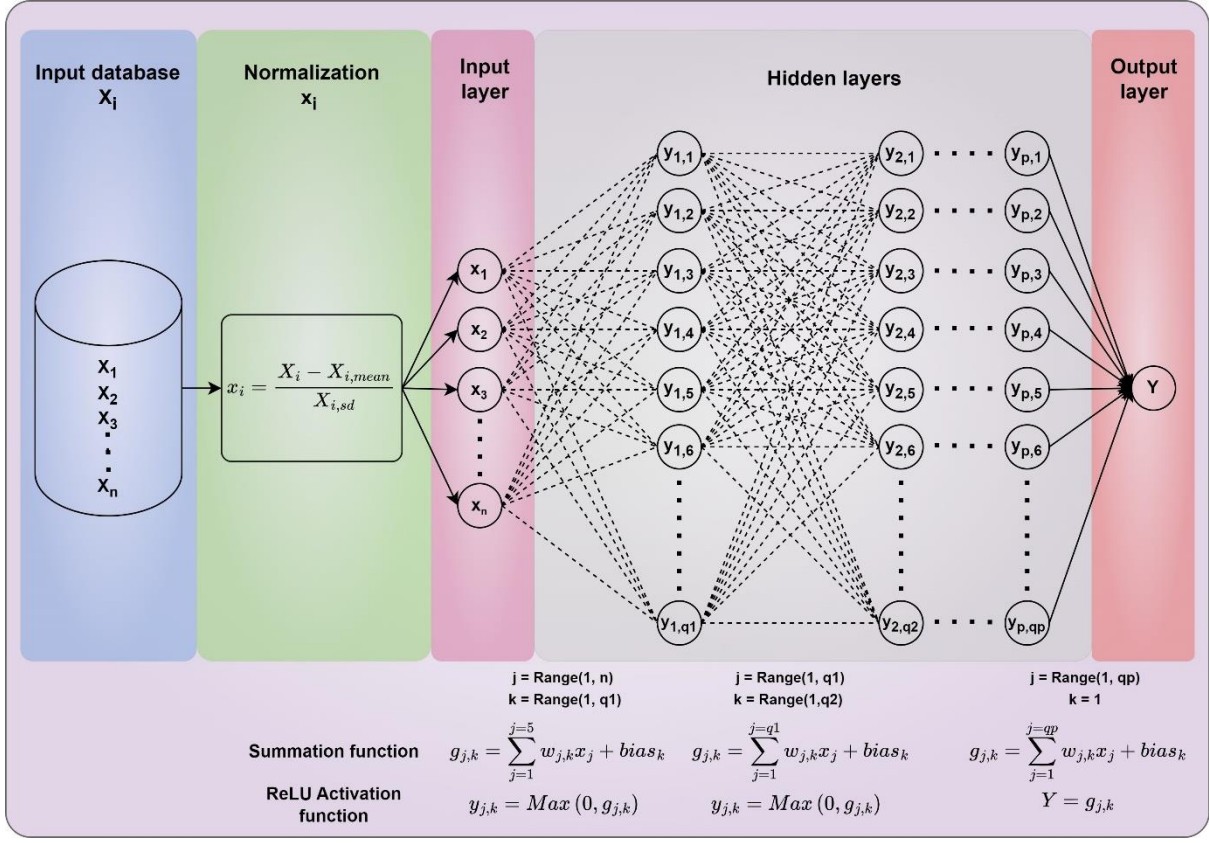


**Figure 3.** Schematic of the ANN architecture employed in the modelling of TSL and OHC parameters. The flow of the modelling and the associated mathematical transformations/formulations are given by considering a typical ANN architecture with *n* input parameters, one output parameter, *p* hidden layers, and *q1* to *qp* neurons in each hidden layer.

## 4. Results and discussion

The performance of TSL and OHC models on unseen data from the in-situ and satellite sources was assessed using density
scattergrams and statistical metrics. These metrics include mean bias error (MBE), mean bias percentage error (MBPE), mean
absolute error (MAE), mean absolute percentage error (MAPE), root mean square error (RMSE), Pearson correlation
coefficient (R), slope, and intercept (also referred and presented in Prakash and Shanmugam, 2022). To better understand the
model performance, mean values of in-situ data were computed for the validation period and used to compute the weighted
average of validation metrics across all the depth extents.





## 4.1. Validation with independent in-situ data

The main objective of the independent validation using in-situ data is to evaluate the generalization ability and overall accuracy of TSL and OHC-ANN models on unseen data. For this purpose, the in-situ measured variables such as SST, SSS, and latitude / longitude were inputted into these models to output the predicted values which were then compared with in-situ TSL and OHC data. The number of independent validation data points and their spatial distribution are presented in Table 2 and Fig. A1(b). The results in the form of density scattergrams are shown in Figs. 4 and 5. These results showed high correlation and low errors with the model-predicted values. From Table 2 and Fig. 4, the performance of the TSL models is exceptionally good on unseen data of all the depth extents without any overfitting. Similar model performance can also be observed in the case of OHC estimates as it primarily depends on the TSL estimates (Table 2 and Fig. 5). The high values of R indicate a strong positive correlation between the predicted and in-situ OHC (TSL) values. This suggests that the models are generally capable of capturing OHC (TSL) patterns in the data. The slope and intercept of the regression line between predicted and actual values are close to 1 and 0, respectively. This suggests that the model-predicted values have good agreement with the actual values with a minimal bias. The RMSE values are notably small implying that the predicted OHC values have a little random error when compared to the actual data. The MBE and MBPE values are close to zero, indicating that the model-predicted values have a negligible systematic error when compared to the actual values. The low MAE and MAPE values are also indicating a high accuracy with the model-predicted OHC values. These results clearly demonstrate that the proposed ANN models succeeded in generalizing and accurately predicting the measured OHC (TSL) data with a high accuracy.

Spatial distribution of mean percentage error (MPE) over the global open oceanic region was computed by averaging the observed percentage errors of all modelling depths available at each pixel (Fig. A2) for estimating the OHC changes. It is observed that the models' performance is comparatively low over the north-western parts of the North Atlantic gyre, southwestern parts of the South Atlantic gyre, Kuroshio extension, and Antarctic circumpolar regions. An elaborate note on the potential sources of the observed MPE values is given in Sect. 4.4. Further, the entire validation dataset was divided into two parts in terms of the observed overestimation and underestimation of data. In the cases of overestimation (underestimation), 95% of the data points have a MPE of less than or equal to 0.47% (0.44%). The lower values of MPE indicate that the proposed ANN models succeed in capturing the OHC patterns in all major oceanic basins and can be used to produce accurate OHC products based on their implementation on real-time data.





**Figure 4.** Density scatterplots showing the observed agreement between model-predicted TSL values and in-situ measured TSL values during insitu-based independent validation.





**Figure 5.** Density scatterplots showing the observed agreement between model-predicted OHC values and in-situ measured OHC values during insitu-based independent validation.





**Table 2.** Statistical results from the insitu-based independent validation data of TSL (regular font) and OHC (bold font) against unseen Argo measured in-situ data. The units for the various metrics used in TSL & OHC validations are given as follows: Mean (m & GJ m$^{-2}$), RMSE (m & GJ m$^{-2}$), MBE (m & GJ m$^{-2}$), MBPE (%), MAE (m & GJ m$^{-2}$), MAPE (%), and intercept (m & GJ m$^{-2}$).

| Depth (m) | N Data for model development | N Data for independent validation | Mean | R | RMSE | MBE | MBPE | MAE | MAPE | Slope | Intercept |
|---|---|---|---|---|---|---|---|---|---|---|---|
| 20 | 801303 | 536719 | 1.44 | 0.9997 | 0.01 | -0.0007 | 0.0575 | 0.006 | 0.60 | 0.9981 | 0.002 |
|  |  |  | **23.91** | **0.9997** | **0.02** | **-0.0011** | **-0.0047** | **0.009** | **0.04** | **0.9987** | **0.030** |
| 30 | 794166 | 532149 | 2.15 | 0.9993 | 0.03 | 0.0029 | 0.3764 | 0.015 | 0.99 | 0.9982 | 0.007 |
|  |  |  | **32.85** | **0.9992** | **0.04** | **0.0010** | **0.0027** | **0.021** | **0.06** | **0.9992** | **0.030** |
| 40 | 787074 | 526571 | 2.85 | 0.9988 | 0.05 | -0.0009 | 0.1325 | 0.027 | 1.28 | 0.9988 | 0.002 |
|  |  |  | **47.78** | **0.9988** | **0.07** | **-0.0008** | **-0.0014** | **0.038** | **0.08** | **0.9978** | **0.103** |
| 50 | 779134 | 520102 | 3.54 | 0.9984 | 0.07 | -0.0008 | 0.0861 | 0.042 | 1.47 | 0.9975 | 0.008 |
|  |  |  | **59.70** | **0.9984** | **0.10** | **0.0015** | **0.0028** | **0.057** | **0.10** | **0.9972** | **0.169** |
| 100 | 731065 | 476709 | 6.80 | 0.9974 | 0.18 | -0.0129 | -0.1725 | 0.120 | 2.09 | 0.9960 | 0.015 |
|  |  |  | **119.00** | **0.9973** | **0.25** | **-0.0279** | **-0.0233** | **0.169** | **0.14** | **0.9981** | **0.196** |
| 150 | 712120 | 460278 | 9.83 | 0.9967 | 0.29 | -0.0407 | -0.3419 | 0.205 | 2.41 | 0.9905 | 0.053 |
|  |  |  | **177.97** | **0.9965** | **0.40** | **-0.0369** | **-0.0198** | **0.279** | **0.16** | **0.9867** | **2.331** |
| 200 | 697314 | 446979 | 12.64 | 0.9961 | 0.38 | -0.0001 | 0.0571 | 0.272 | 2.51 | 0.9960 | 0.050 |
|  |  |  | **236.62** | **0.9959** | **0.53** | **-0.0076** | **-0.0029** | **0.372** | **0.16** | **0.9939** | **1.426** |
| 250 | 686378 | 436906 | 15.28 | 0.9959 | 0.46 | -0.0361 | -0.1803 | 0.332 | 2.49 | 0.9943 | 0.051 |
|  |  |  | **295.04** | **0.9957** | **0.63** | **-0.0242** | **-0.0078** | **0.450** | **0.15** | **0.9918** | **2.392** |
| 300 | 678526 | 429501 | 17.80 | 0.9956 | 0.55 | -0.0471 | -0.0023 | 0.392 | 2.53 | 0.9851 | 0.218 |
|  |  |  | **353.29** | **0.9954** | **0.74** | **-0.0155** | **-0.0039** | **0.525** | **0.15** | **0.9889** | **3.902** |
| 350 | 672148 | 423688 | 20.23 | 0.9949 | 0.65 | -0.1035 | -0.3383 | 0.462 | 2.59 | 0.9860 | 0.179 |
|  |  |  | **411.40** | **0.9947** | **0.87** | **-0.0357** | **-0.0081** | **0.613** | **0.15** | **0.9861** | **5.676** |
| 400 | 666605 | 418686 | 22.57 | 0.9947 | 0.72 | -0.0425 | -0.0526 | 0.505 | 2.52 | 0.9887 | 0.213 |
|  |  |  | **469.39** | **0.9945** | **0.97** | **-0.0067** | **-0.0010** | **0.676** | **0.14** | **0.9879** | **5.683** |
| 450 | 661336 | 413987 | 24.83 | 0.9946 | 0.78 | -0.1227 | -0.4726 | 0.547 | 2.47 | 0.9916 | 0.087 |
|  |  |  | **527.25** | **0.9943** | **1.06** | **-0.1681** | **-0.0315** | **0.741** | **0.14** | **0.9872** | **6.588** |
| 500 | 654880 | 408240 | 27.03 | 0.9949 | 0.80 | -0.0604 | -0.1866 | 0.558 | 2.29 | 0.9945 | 0.089 |
|  |  |  | **585.03** | **0.9947** | **1.07** | **-0.0761** | **-0.0127** | **0.747** | **0.13** | **0.9894** | **6.105** |
| 550 | 649850 | 403357 | 29.14 | 0.9948 | 0.85 | -0.0462 | -0.0937 | 0.586 | 2.19 | 0.9911 | 0.213 |
|  |  |  | **642.69** | **0.9945** | **1.15** | **0.0347** | **0.0057** | **0.787** | **0.12** | **0.9900** | **6.479** |
| 600 | 645150 | 398855 | 31.21 | 0.9945 | 0.91 | -0.0390 | -0.0205 | 0.623 | 2.18 | 0.9883 | 0.327 |
|  |  |  | **700.28** | **0.9942** | **1.23** | **0.0298** | **0.0046** | **0.838** | **0.12** | **0.9873** | **8.937** |
| 650 | 640479 | 392921 | 33.18 | 0.9941 | 0.99 | 0.0185 | 0.0903 | 0.670 | 2.19 | 0.9949 | 0.189 |
|  |  |  | **757.74** | **0.9939** | **1.33** | **0.0086** | **0.0014** | **0.892** | **0.12** | **0.9904** | **7.296** |
| 700 | 633004 | 388469 | 35.13 | 0.9941 | 1.04 | -0.1928 | -0.4791 | 0.711 | 2.17 | 0.9858 | 0.307 |
|  |  |  | **815.15** | **0.9938** | **1.41** | **-0.2413** | **-0.0292** | **0.960** | **0.12** | **0.9836** | **13.134** |
| **Weighted average** |  |  | 0.9961 | 0.74 | -0.0620 | -0.1591 | 0.513 | 2.29 | 0.9927 | 0.177 |  |
|  |  |  | **0.9960** | **1.03** | **-0.0515** | **-0.0087** | **0.708** | **0.13** | **0.9914** | **6.648** |  |





## 4.2. Satellite-based independent validation

The performance of the proposed ANN models in satellite-based applications has been assessed by injecting daily SST and SSS data from the satellite sources in place of the in-situ sources. The choice of satellite sources for SST and SSS data is completely subjective to the intended application and their compatibility in terms of spatial and temporal resolutions, whereas geographical coordinates data can be employed from WOA corresponding to the climatological TSL and OHC data. It is recommended to resample SST and SSS data to the WOA grid to eliminate the discrepancies arising from the non-uniform spatial references among the input data. In the current study, the NOAA Advanced Very High-Resolution Radiometer (AVHRR) Optimum Interpolation Sea Surface Temperature products (OISST v2.1) were used for daily SST data of 0.25° spatial resolution (Huang et al., 2021). Daily SSS data of the same spatial resolution were obtained from the ORAS5 reanalysis system of the European Centre for Medium-Range Weather Forecasts at the CMEMS portal (Product ID: GLOBAL_REANALYSIS_PHY_001_031) (Zuo et al., 2017). The NetCDF4 and NumPy Python libraries were used to read and resample satellite data to the WOA-18 grid, and to collocate with the corresponding Argo in-situ data points. The accuracy of the satellite-based SST and SSS was verified by Argo-measured SST and SSS profile data (N = 244722). The observed R, RMSE, MBE, and MAE values in SST & SSS validations are 0.99 & 0.99, 0.51°C & 0.26 PSU, -0.05°C & -0.006 PSU, and 0.33°C & 0.12 PSU, respectively. High correlation coefficients and low errors indicate the minimal deviation of satellite-based data from actual (in-situ) data and ensure the reliability of satellite data in accurately representing the physical oceanographic conditions. The satellite-based SST, SSS, latitude, and longitude data were then given as the inputs to the ANN models for producing TSL and OHC estimates of all the depth extents considered in this study. Consequently, the model-derived TSL and OHC estimates were compared with Argo-measured in-situ data, and the satellite-based independent validation results are presented in this section (Table 3 and Figs. 6 and 7).

The performance of the proposed ANN models on satellite-based independent validation data (Table 3, Figs. 6 and 7) is rather similar to their performance on in-situ-based independent validation data (Table 2, Figs. 4 and 5). However, the models' performance on satellite-based independent validation data was marginally low as compared to the in-situ-based validation data, likely due to the errors associated with the satellite-derived products. According to the statistical results, the R values were observed to be slightly lower by an average percentage decrease of 0.11% across all depth extents. Similarly, the RMSE, MBE, MBPE, MAE, and MAPE were slightly larger than those values observed during the in-situ-based independent validation datasets. This relatively lower performance of the proposed models on the satellite-based independent validation datasets can be observed by comparing the spatial maps and the distribution of MPE (Figs. A2 and A3). The relatively higher magnitudes of MPE can be observed over the northwestern parts of the North Atlantic gyre, southwestern parts of the South Atlantic gyre, Kuroshio extension, and Antarctic circumpolar regions based on in-situ-based validation data. And, 95% of the data have a MPE of less than or equal to 0.56% (0.5%) in the cases of overestimation (underestimation), which is higher than those reported in Sect. 4.1. Though the performance of the proposed models' on satellite-based data is comparatively lower than the in-situ-based validation data, the observed difference in various validation metrics is rather insignificant. It



substantiates the efficiency of the proposed models in estimating OHC from satellite data at various depth extents over the major oceanic basins. However, it should be noted that the validation results presented in this section are subject to vary with the other sources of satellite-based SST and SSS data.

**Figure 6.** Density scatterplots showing the observed agreement between model-predicted TSL values and in-situ measured TSL values during satellite-based independent validation.





**Figure 7.** Density scatterplots showing the observed agreement between model-predicted OHC values and in-situ measured OHC values during satellite-based independent validation.



**Table 3.** Statistical results from satellite-based independent validation data of TSL (regular font) and OHC (bold font) against unseen Argo measured in-situ data. The units for the various metrics used in TSL & OHC validations are given as follows: Mean (m & GJ m⁻²), RMSE (m & GJ m⁻²), MBE (m & GJ m⁻²), MBPE (%), MAE (m & GJ m⁻²), MAPE (%), and intercept (m & GJ m⁻²).

| Depth (m) | N Data for model development | Data for independent validation | Mean | R | RMSE | MBE | MBPE | MAE | MAPE | Slope | Intercept |
|---|---|---|---|---|---|---|---|---|---|---|---|
| **20** | 801303 | 536719 | 1.44 | 0.9987 | 0.03 | -0.0034 | -0.0822 | 0.016 | 1.67 | 0.9960 | 0.002 |
| | | | **23.91** | **0.9987** | **0.04** | **-0.0049** | **-0.0201** | **0.023** | **0.09** | **0.9965** | **0.080** |
| **30** | 794166 | 532149 | 2.15 | 0.9984 | 0.04 | -0.0008 | 0.2562 | 0.027 | 1.88 | 0.9961 | 0.008 |
| | | | **32.85** | **0.9984** | **0.06** | **-0.0043** | **-0.0118** | **0.037** | **0.10** | **0.9969** | **0.108** |
| **40** | 787074 | 526571 | 2.85 | 0.9980 | 0.07 | -0.0054 | 0.0211 | 0.041 | 2.08 | 0.9969 | 0.003 |
| | | | **47.78** | **0.9980** | **0.09** | **-0.0070** | **-0.0143** | **0.057** | **0.12** | **0.9959** | **0.191** |
| **50** | 779134 | 520102 | 3.54 | 0.9977 | 0.09 | -0.0060 | -0.0262 | 0.057 | 2.17 | 0.9960 | 0.008 |
| | | | **59.70** | **0.9976** | **0.12** | **-0.0056** | **-0.0090** | **0.077** | **0.13** | **0.9956** | **0.257** |
| **100** | 731065 | 476709 | 6.80 | 0.9966 | 0.20 | -0.0206 | -0.2651 | 0.140 | 2.56 | 0.9951 | 0.013 |
| | | | **119.00** | **0.9965** | **0.28** | **-0.0385** | **-0.0322** | **0.194** | **0.16** | **0.9971** | **0.301** |
| **150** | 712120 | 460278 | 9.83 | 0.9958 | 0.32 | -0.0496 | -0.4165 | 0.229 | 2.81 | 0.9897 | 0.052 |
| | | | **177.97** | **0.9956** | **0.44** | **-0.0491** | **-0.0266** | **0.311** | **0.17** | **0.9858** | **2.474** |
| **200** | 697314 | 446979 | 12.64 | 0.9951 | 0.43 | -0.0091 | -0.0022 | 0.300 | 2.83 | 0.9951 | 0.053 |
| | | | **236.62** | **0.9950** | **0.59** | **-0.0200** | **-0.0081** | **0.409** | **0.17** | **0.9929** | **1.653** |
| **250** | 686378 | 436906 | 15.28 | 0.9948 | 0.52 | -0.0450 | -0.2117 | 0.364 | 2.79 | 0.9928 | 0.065 |
| | | | **295.04** | **0.9946** | **0.71** | **-0.0365** | **-0.0119** | **0.492** | **0.17** | **0.9904** | **2.807** |
| **300** | 678526 | 429501 | 17.80 | 0.9943 | 0.62 | -0.0556 | -0.0279 | 0.428 | 2.79 | 0.9837 | 0.235 |
| | | | **353.29** | **0.9941** | **0.83** | **-0.0271** | **-0.0071** | **0.571** | **0.16** | **0.9875** | **4.398** |
| **350** | 672148 | 423688 | 20.23 | 0.9939 | 0.71 | -0.1052 | -0.3291 | 0.494 | 2.80 | 0.9846 | 0.206 |
| | | | **411.40** | **0.9936** | **0.95** | **-0.0381** | **-0.0086** | **0.655** | **0.16** | **0.9847** | **6.264** |
| **400** | 666605 | 418686 | 22.57 | 0.9935 | 0.79 | -0.0450 | -0.0422 | 0.540 | 2.72 | 0.9869 | 0.252 |
| | | | **469.39** | **0.9933** | **1.06** | **-0.0103** | **-0.0017** | **0.723** | **0.15** | **0.9860** | **6.557** |
| **450** | 661336 | 413987 | 24.83 | 0.9934 | 0.87 | -0.1234 | -0.4559 | 0.586 | 2.67 | 0.9898 | 0.129 |
| | | | **527.25** | **0.9931** | **1.17** | **-0.1694** | **-0.0316** | **0.792** | **0.15** | **0.9854** | **7.508** |
| **500** | 654880 | 408240 | 27.03 | 0.9934 | 0.91 | -0.0707 | -0.2034 | 0.605 | 2.50 | 0.9924 | 0.134 |
| | | | **585.03** | **0.9933** | **1.21** | **-0.0909** | **-0.0151** | **0.807** | **0.14** | **0.9874** | **7.293** |
| **550** | 649850 | 403357 | 29.14 | 0.9932 | 0.97 | -0.0484 | -0.0768 | 0.636 | 2.40 | 0.9887 | 0.280 |
| | | | **642.69** | **0.9929** | **1.30** | **0.0315** | **0.0053** | **0.851** | **0.13** | **0.9876** | **8.021** |
| **600** | 645150 | 398855 | 31.21 | 0.9930 | 1.03 | -0.0431 | -0.0139 | 0.675 | 2.38 | 0.9861 | 0.392 |
| | | | **700.28** | **0.9927** | **1.39** | **0.0242** | **0.0039** | **0.906** | **0.13** | **0.9850** | **10.52** |
| **650** | 640479 | 392921 | 33.18 | 0.9926 | 1.11 | 0.0193 | 0.1132 | 0.719 | 2.37 | 0.9925 | 0.267 |
| | | | **757.74** | **0.9924** | **1.48** | **0.0092** | **0.0015** | **0.957** | **0.13** | **0.9880** | **9.090** |
| **700** | 633004 | 388469 | 35.13 | 0.9926 | 1.16 | -0.1917 | -0.4560 | 0.763 | 2.34 | 0.9835 | 0.387 |
| | | | **815.15** | **0.9922** | **1.56** | **-0.2400** | **-0.0290** | **1.029** | **0.13** | **0.9813** | **14.982** |
| **Weighted average** | | | | 0.9950 | 0.83 | -0.0657 | -0.1645 | 0.554 | 2.54 | 0.9909 | 0.224 |
| | | | | **0.9948** | **1.15** | **-0.0566** | **-0.0104** | **0.763** | **0.14** | **0.9896** | **7.799** |





### 4.3. Comparison with the contemporary satellite-based OHC models

Comparison of our ANN models with the existing models is crucial to determine the relative uncertainty in the OHC estimates.
Previously, an ANN algorithm suite was developed by the National Remote Sensing Centre (NRSC) of ISRO to disseminate
the daily OHC products over the North Indian Ocean (40°E-120°E, 0°-30°N) at a spatial resolution of 0.25 degree (Ali et al.,
2012; Jagadeesh et al., 2015). This algorithm suite includes ANN models to estimate OHC at multiple depth extents such as
50 m, 100 m, 150 m, 200 m, 300 m, 500 m, and 700 m for the given input data of sea level anomaly (SLA), SST, and $OHC_{clim,d}$.
It estimates OHC changes by utilizing the satellite altimetry-based SLA data from AVISO (Archiving, Validation, and
Interpretation of Satellite Oceanographic data) data portal, SST from the Advanced Microwave Scanning Radiometer-2
(AMSR2) onboard JAXA's Global Change Observation Mission 1st-Water (GCOM-W1), and climatological OHC from the
World Ocean Atlas-2009 monthly climatological CTD profiles. The multilayer perceptron regressor algorithm of neural
networks with three hidden layers was used to estimate OHC of all seven depth extents. The number of data points used to
develop and validate the NRSC-ANN algorithm were 11472 and 2479, respectively. To estimate OHC changes at different
depths, this algorithm employs the Celsius scale, in-situ temperature, and average density data instead of the Kelvin scale,
conservative temperature, and instantaneous density, respectively (see Eq. 3 in Jagadeesh et al., 2015).
For this inter-comparison purpose, validation datasets were prepared for the period of 2017-2020 by calculating in-
situ OHC in both Kelvin and Celsius scales for the depth extents of 50 m, 100 m, 150 m, 200 m, 300 m, 500 m, and 700 m.
Daily OHC data were obtained from the NRSC's Bhuvan portal and collocated with the corresponding Celsius-scaled in-situ
OHC data to evaluate the NRSC-ANN model products. Similarly, satellite-based SST and SSS data, and climatological TSL
and OHC data were extracted by collocating with Kelvin-scaled in-situ OHC data for our ANN model to generate the OHC
products. Evaluation of these two OHC products was done separately by means of the normalized metrics such as R, MBPE,
and MAPE (Table 4).
**Table 4.** Statistical results for our ANN model and NRSC-ANN model obtained from another independent dataset of different
depth extents used in this study.

| Depth (m) | N | R | | MBPE (%) | | MAPE (%) | |
|---|---|---|---|---|---|---|---|
| | | NRSC-ANN model | Proposed ANN model | NRSC-ANN model | Proposed ANN model | NRSC-ANN model | Proposed ANN model |
| **50** | 15595 | 0.9223 | 0.9303 | -0.0012 | 0.0227 | 1.4762 | 0.1104 |
| **100** | 14546 | 0.8575 | 0.8780 | -0.3539 | 0.0303 | 2.5145 | 0.1732 |
| **150** | 14303 | 0.7678 | 0.8215 | -0.6887 | -0.0263 | 3.2401 | 0.2053 |
| **200** | 13513 | 0.7169 | 0.8152 | -1.1048 | 0.0072 | 3.4667 | 0.1903 |
| **300** | 12833 | 0.7732 | 0.8690 | -1.2656 | 0.0218 | 3.1671 | 0.1525 |
| **500** | 12410 | 0.8965 | 0.9346 | -0.6996 | -0.0052 | 2.3939 | 0.1073 |
| **700** | 11959 | 0.9447 | 0.9628 | -0.6214 | -0.0370 | 2.0035 | 0.0891 |



As expected, our ANN model gave more accurate OHC estimates for all depth extents and hence yielded higher

correlation coefficients and lower errors as compared to the NRSC-ANN model. The accuracy of OHC estimates produced by
our ANN model also increased with depth in contrast to that of NRSC-ANN OHC estimates. Our ANN model was
accomplished with the selection of key input parameters based on a precise theoretical basis, accurate computation of in-situ
parameters, and selection of separate ANN architectures.

It should be mentioned that SLA is the combined outcome of temperature (thermosteric), salinity (halosteric), and

water mass changes in the oceanic water column. The direct use of satellite altimeter-derived SLA without eliminating
halosteric and water mass change components results in weaker correlations with OHC of various depth extents. Moreover,
the different time spans were used in the computation of the mean sea level at AVISO (1993-2012) and monthly climatology
data at WOA09 (1955-2006). The pair of merged SLA data from AVISO/CMEMS and climatological OHC data from WOA
could lead to discrepancies in OHC estimates. Under these considerations, the present study was focused on TSL modelling
rather than the Alt-GRACE approach (Meyssignac et al., 2019) to implement the ocean thermal expansion method to estimate
OHC changes.

Celsius scale can be used to compute in-situ OHC where the temperature gradient is always on the positive side. The

usage of the Celsius scale when the temperatures are less than zero and greater than the seawater freezing point is not
appropriate because of the potential negative values. In addition, the conservative temperature is an accurate variable compared
to the direct in-situ temperature or potential temperature. It represents the actual heat content of a mixture of two water masses
which are characterized by variations of salinity, pressure, and temperature (Pawlowicz, 2013). Thus, the 9conservative
temperature is defined in absolute scale (Kelvin scale) and used to calculate the in-situ OHC. On the other hand, employing
instantaneous density rather than average density is essential to account for the variations in seawater density which is
determined by temperature and salinity changes.

The vertical distribution of conservative temperature varies from equatorial to polar regions, and it follows a non-

linear profile with a mixed layer at the top, a thermocline at the middle, and a deep ocean layer at the bottom. This suggests
that it is appropriate to customize the ANN hyperparameters for each modelling depth. In this study, hyperparameter tuning
was performed for each modelling depth and it resulted in a better understanding of OHC patterns at various depth extents.
Though a clear improvement was achieved with the proposed OHC models, a relatively lower correlation was observed for
our ANN models in the depth range of 100-300 m over the North Indian Ocean (refer to Table 4). Similar results were obtained
for the NRSC-ANN models as well. It implies that the proposed ANN models less generalized the OHC patterns at the
intermediate depths over the North Indian Ocean. The underlying factors for the less generalized OHC patterns are described
in the following section. Nevertheless, the results demonstrated that the proposed ANN models contributed to improving the
accuracy and quality of OHC products through the ocean thermal expansion method.



### 4.4. Potential sources of uncertainty in OHC estimates

The relationship between the surficial parameters (SST and SSS) and depth-integrated parameters (TSL and OHC) is the prime factor for determining the efficiency of the proposed OHC models of various depth extents (Klemas and Yan, 2014). This relationship is mainly influenced by a wide range of geophysical processes including ocean currents, vertical mixing (upwelling/downwelling), stratification, fronts, gyres, eddies, and air-sea interface processes. In addition, different climate modes and oscillations, solar radiation, sea ice, phytoplankton growth, freshwater inputs, and winds can also be considered in this context. Monthly climatological CTD profiles obtained from the WOA-18 database were objectively analyzed to calculate the mean SST and SSS fields over a period of 1955-2017. Hence, these climatological data along with real-time SST and SSS data enabled the ANN models to better generalize the prevailing geophysical processes and subsequent patterns in TSL & OHC of various depth extents. The same can be perceived from the improved accuracy levels observed during the independent validations carried out on unseen data (refer to Sects. 4.1 and 4.2) and the comparison with NRSC-OHC model products (Sect. 4.3).

It should be noted that the established relationship between the input parameters (surficial and climatological) and output parameters (TSL & OHC patterns) may not hold great in the events of the above complex geophysical processes where the physical oceanographic conditions differ significantly from the prevailing conditions. Moreover, the relative contributions of these geophysical processes are subject to vary depending on the time and location of the water parcel in oceans. Slightly lower accuracy of the proposed ANN models can be attributed to the influence of these complex geophysical processes. The in-situ and satellite-based retrieval of all these atmospheric/surface/subsurface processes and their incorporation into the ANN models is difficult because of the scarcity/sparsity of the required datasets in different spatial, temporal, and vertical scales. The above factors constitute a potential source of uncertainty in OHC estimates and reduce the generalization ability of the model. Further efforts are needed to better understand, quantify, and eliminate the different sources of observed uncertainties caused by the complex geophysical oceanic processes. More number of in-situ CTD profiles are required to be collected and analyzed in such oceanic regions to address the associated complex patterns and processes. Future releases of WOA will certainly resolve the complex patterns in OHC data with the inclusion of newly collected CTD profiles over various oceanic basins.

### 5. A preliminary analysis of global OHC data

This section presents a map of the global distribution of model-derived OHC estimates and its variation from 1993 to 2020. This time period (1993-2020) was chosen based on the availability of satellite-based input data. The NOAA's AVHRR OISST version 2.1 products are available from 1981 to present, but the ECMWF's ORAS5 SSS data are available from 1993 to 2020 only. Thus, for this preliminary analysis, daily OHC estimates were generated for the period from 1993 to 2020 and the corresponding annual mean estimates were computed as shown in Figs. 8a and 8b. Subsequently, the OHC anomalies (OHCA) were estimated by subtracting the annual mean data of 2020 from those of 1993 (Fig. 8c). It is worth mentioning that the heat



content estimates presented in this section cover all the modelling depths. The bathymetry values of each pixel were rounded
off to the nearest and lowest modelling depth (d) with the help of GEBCO-2020 bathymetry data (GEBCO Compilation Group,
2020), and the corresponding $OHC_d$ values were considered for that pixel. As the proposed models are built for open oceanic
regions, the regions covered by sea ice are masked in both north and south poles by verifying the corresponding daily sea ice
concentration data obtained from the National Snow and Ice Data Center (Meier et al., 2021).

The spatial patterns of annual OHC product are nearly identical in both 1993 and 2020 (Figs. 8a and 8b). The relatively

higher OHC values (830 GJ m$^{-2}$ - 840 GJ m$^{-2}$) at a depth of 700 m are observed over the North Atlantic subtropical gyre, Gulf
stream, north Arabian sea, eastern Mediterranean sea, and southeastern side of the Madagascar. Next, the OHC values of range
820 GJ m$^{-2}$ - 830 GJ m$^{-2}$ are observed over a vast portion of the major oceanic basins, namely the western parts of the north
Pacific ocean (105° E – 150° W, 0° – 35° N), the western and central parts of the south Pacific ocean (115° E – 100° W, 0° –
40° S), the north Atlantic ocean (90° W – 30° E, 0° – 40° N, excluding the north Atlantic subtropical gyre, the Gulf stream,
and the eastern Mediterranean sea), the southwestern half of the south Atlantic ocean (55° W – 15° E, 15° S – 40° S), and the
Indian ocean (25° E – 150° E, 25° N - 40° S, excluding the north Arabian sea and the southeastern side of the Madagascar).
Subsequently, the OHC values in the range of 810 GJ m$^{-2}$ - 820 GJ m$^{-2}$ are observed over the eastern part of the Pacific ocean
and equatorial Pacific ocean (170° E– 70° W, 45° S – 40° N), northeastern part of the north Atlantic ocean (50° W – 5° W,
40° – 55° N) , the equatorial Atlantic ocean, the northeastern half of the south Atlantic ocean (60° W – 25° E, 0° S – 45° S),
and the southeastern parts of the Indian ocean (60° E – 150° E, 30° S – 50° S). And, the OHC values in the range of 780 GJ
m$^{-2}$ - 810 GJ m$^{-2}$ are observed in the latitudinal range of 40° – 70° in both the hemispheres with a decreasing trend from 40°
to 70°N. In the case of oceanic regions shallower than 700 m, the OHC values are observed to vary widely as shown in Figs.
8a and 8b.

Though the spatial patterns and magnitudes of OHC are almost same during the years 1993 and 2020, well-marked

warming/cooling regions can be observed in the order of $O(10^0)$ during 2020 with reference to 1993 (Fig. 8c). The ocean
warming of 1.5 GJ m$^{-2}$ - 5 GJ m$^{-2}$ is observed over the Red Sea, Kuroshio extension, Gulf Stream, and southern side of the
Africa (30° S – 45° S). On the other hand, cooling of the same range is observed over the north Atlantic subpolar gyre and
southern parts of the Indian Ocean. In addition, several eddy regions are also seen with warming/cooling patterns (Fig. 8c).

In a broader perspective, the observed spatial patterns in OHC and its temporal variation can be attributed to

consistent/anomalous warming/cooling in response to the ocean / atmospheric circulations, anthropogenic climate change and
internal variability. Consequently, a comprehensive analysis is to be carried out to gain new insights into the underlying
phenomena responsible for the observed OHC distributions and its variation in a separate study.




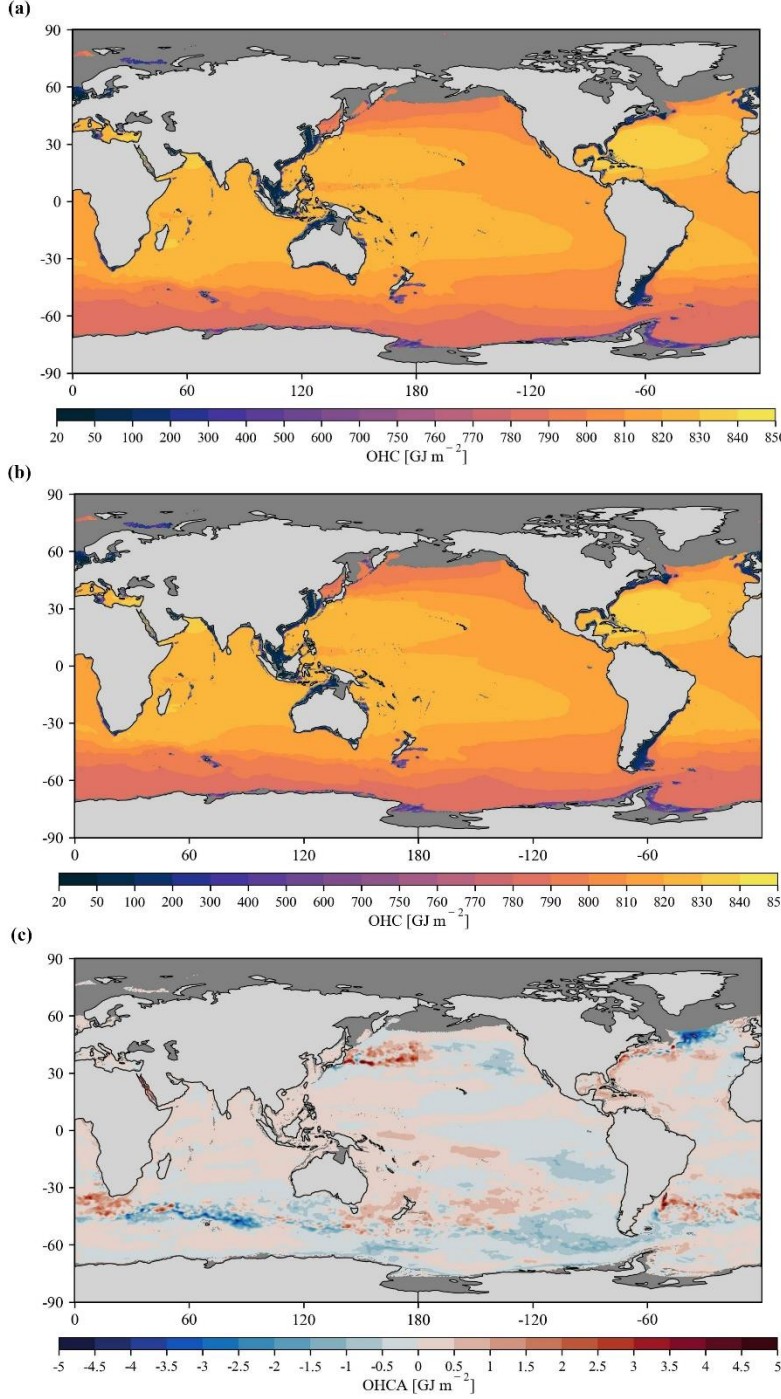

**Figure 8.** Maps showing the spatial distribution of OHC during (a) 1993 and (b) 2020, and (c) the observed variation in 2020 with reference to 1993. The oceanic regions shallower than 20 m depth and/or covered with sea ice are masked with a dark gray flag.





## 6. Conclusion

Accurate reconstruction of OHC and analysis of its regional patterns and long-term global records are critical for estimating the Earth Energy Imbalance and understanding the evolution of the climate change. Owing to the lack of instrumentation to cover geographic and depth ranges, OHC estimates from the in-situ measured temperatures are temporally limited and insufficiently widespread to capture its spatiotemporal changes and structures. OHC estimates from either different mapping methods or Ocean reanalyses (ORAs) have yielded large uncertainties in past studies. Thus, improving OHC estimates through a novel satellite-based method is the major step forward to overcome sparse observations and reduce the uncertainty in OHC trends. In this study, we proposed an artificial network model to estimate OHC changes in global oceans. The proposed ANN model incorporates the ocean thermal expansion method as a promising tool to estimate OHC changes from satellite data. Accurate implementation of the ocean thermal expansion method was challenging due to the inability of the present-day satellite systems to directly measure the ocean thermal expansion/contraction component. In this study, we proposed a satellite-based novel approach to better implement the ocean thermal expansion method by establishing a relationship between the surficial parameters such as SST & SSS and subsurface T-S profiles. This model predicts the depth-integrated TSL component by making use of SST & SSS data and then utilizes the predicted TSL to estimate OHC changes. For this application, we developed ANN models for TSL and OHC of various depth extents such as 20 m, 30 m, 40 m, 50 m, 100 m, 150 m, 200 m, 250 m, 300 m, 350 m, 400 m, 450 m, 500 m, 550 m, 600 m, 650 m, and 700 m. The performance of these TSL & OHC models was assessed by using in-situ-based independent data and satellite-based independent validation data, which were extracted from the unseen in-situ CTD profiles of the Argo program. Observed high correlations and low errors indicated that the proposed ANN models performed exceptionally good on unseen data of all depth extents without any overfitting and can be used in conjunction with the sea ice thermodynamics-based OHC model of the ice-covered oceanic regions (Prakash and Shanmugam, 2022) to better study the trends and patterns in three-dimensional distribution of OHC in the global oceans.

The model development and validation databases were prepared by using in-situ CTD profiles obtained from the Argo program and collocated with the corresponding satellite-based daily data of SST (AVHRR v2.1) and SSS (ORAS5). The multilayer perceptron regressor algorithm of deep neural networks was used and its architecture was optimized by evaluating different combinations of hyperparameters for each modelling depth using the particle swarm optimization technique. Precise consideration of theoretical aspects in the selection of input parameters, accurate computation of in-situ OHC, and customized ANN architectures enabled the proposed models to establish the accurate relationships between the surficial parameters and depth-integrated OHC (TSL) of various depths extents. The overall performance of the proposed models on satellite data was good, suggesting that these models can be used for a variety of applications subjected to the accuracy requirements and can produce accurate satellite-based OHC (TSL) estimates at various depth extents than previously possible. However, the influence of complex geophysical processes on the generalization ability of ANN models is discussed, and realized that the proposed models relatively less generalized the data in the events of complex geophysical processes. Further research should focus on implementation of these models over the oceanic regions with complex geophysical processes. More number of in-



situ CTD profiles need to be collected and analyzed in such oceanic regions to address the associated complex patterns. Future releases of WOA will certainly resolve the complex patterns in OHC data with the inclusion of newly collected CTD profiles over various oceanic basins. However, the scope of the current research includes minimizing the observed marginal gap by exploring new methods/parametrizations in satellite-based OHC modelling approaches.

**CRediT authorship contribution statement**

**Vijay Prakash Kondeti:** Conceptualization, Data curation, Formal analysis, Funding acquisition, Investigation, Methodology, Software, Validation, Visualization, and Writing - original draft. **Palanisamy Shanmugam:** Conceptualization, Formal analysis, Funding acquisition, Investigation, Methodology, Project administration, Resources, Supervision, and Writing - review & editing.

**Code and Data availability**

Data will be made available on request.

**Declaration of competing interest**

The authors declare no known competing financial or personal interests in this paper.

**Acknowledgement**

This research work was supported by The Prime Minister's Research Fellows (PMRF) Scheme and in part by the National Geospatial Programme (NGP) of Department of Science and Technology of Government of India (Grant No: OEC1819150DSTXPSHA). The authors are thankful to the Argo program for providing in-situ CTD profiles. They are grateful to NOAA for WOD-18, WOA-18, and SST data; CMEMS for SSS data, GEBCO for bathymetry data.

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





## Appendix A

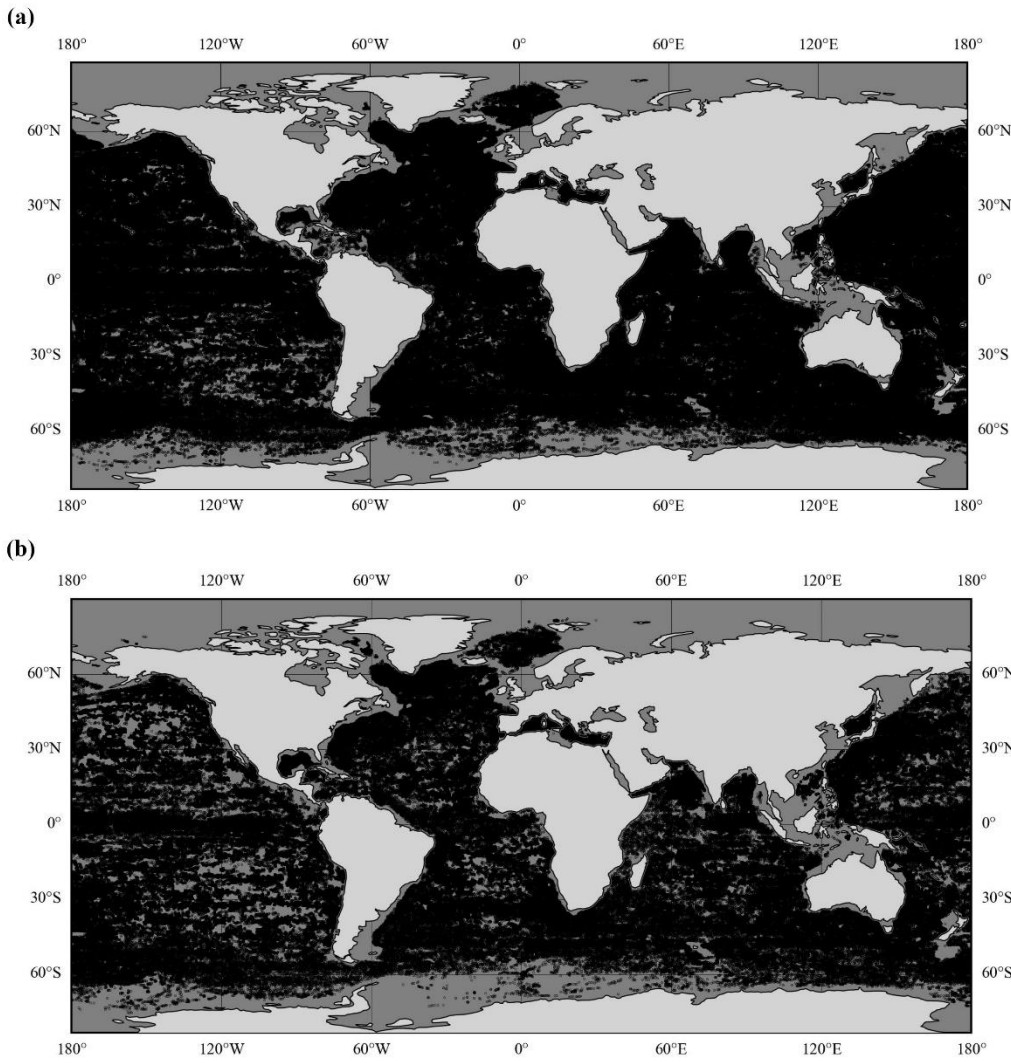

**Figure A1.** The spatial distribution of in-situ data points used for (a) model development (N=633004 Argo CTD profiles) and

(b) validation (N=388469 unseen Argo CTD profiles) in the case of $TSL_{700}$ and $OHC_{700}$.



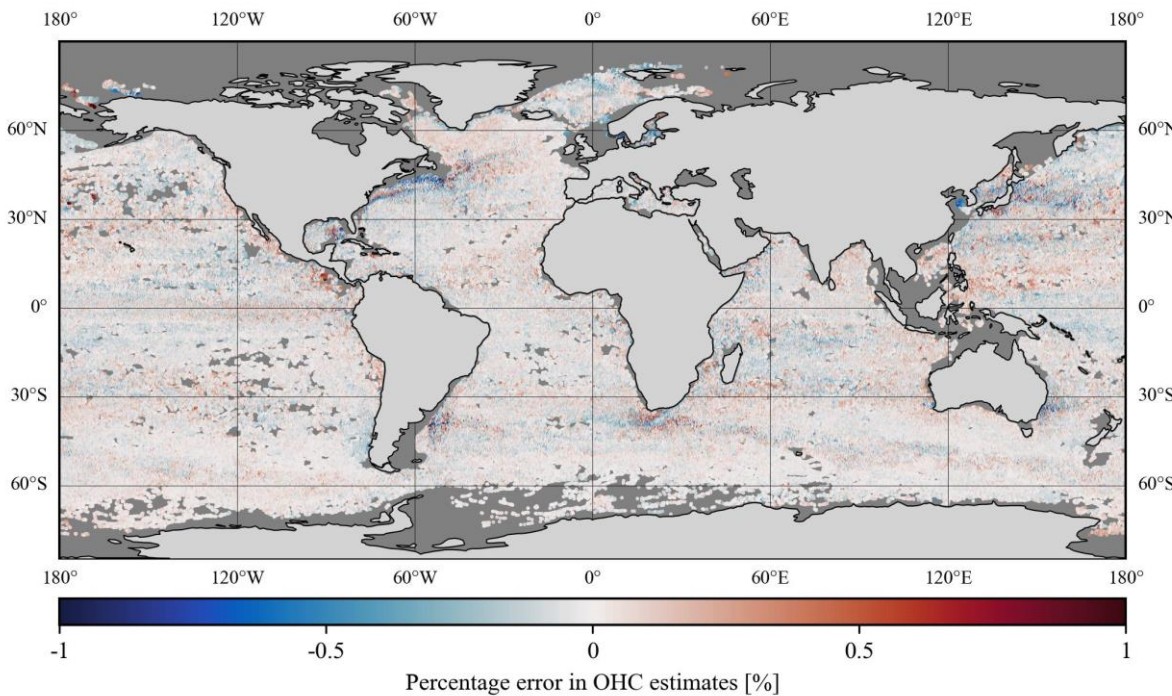

**Figure A2.** Spatial distribution of mean percentage errors observed during the insitu-based independent validation of OHC models. The oceanic regions shallower than 20 m and/or covered with sea ice are marked with a dark gray color.



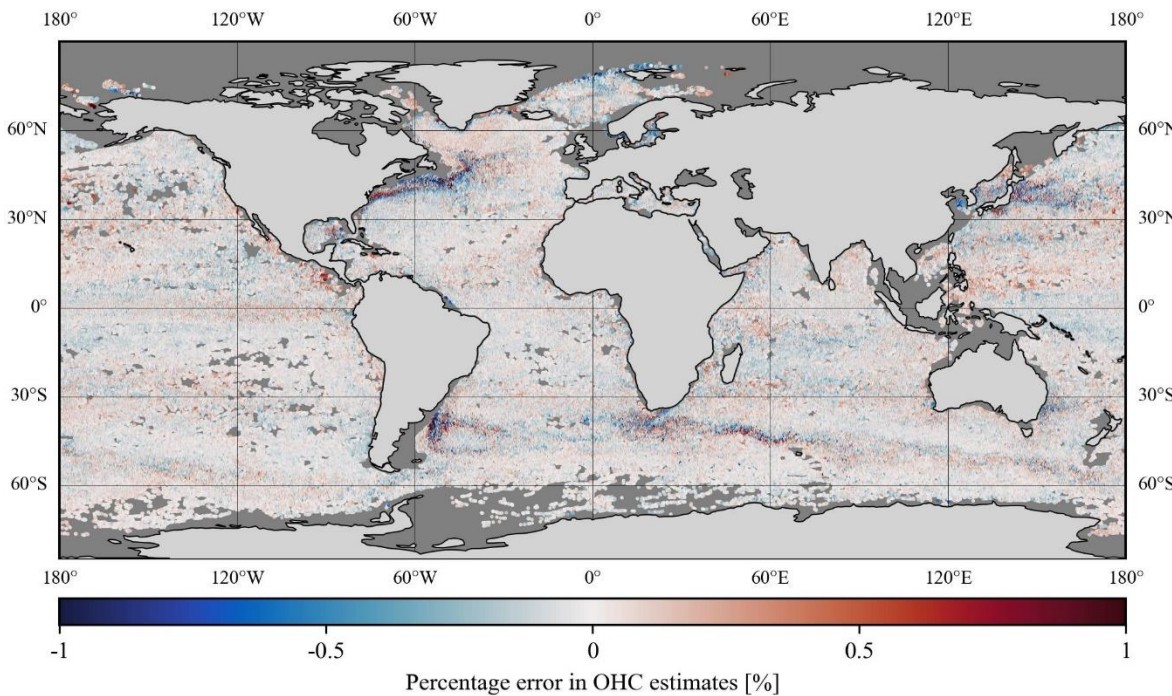

**Figure A3.** Spatial distribution of mean percentage errors observed during the satellite-based independent validation of OHC.

The oceanic regions shallower than 20 m and/or covered with sea ice are marked with a dark gray color.