# Peer review of "Estimating ocean heat content from the ocean thermal expansion parameters using satellite data"

_Earth System Dynamics, 2024_

## Author Comment (AC1)

Review of esd-2024-1, "Estimating ocean heat content from the ocean thermal expansion parameters using satellite data", submitted by Kondeti and Palanisamy for possible publication in Earth System Dynamics.

This manuscript presents a methodology for making maps of ocean heat content (and thermosteric sea level) using an artificial neural network that incorporates Argo CTD data together with what the authors identify as satellite SST and SSS data, as well as the results of the application of that method. There are some very major issues that will need to be resolved prior to publication. Specific minor comments follow, first major, then minor, indexed by line number (L) where possible.

Thank you for your constructive comments that have improved the manuscript. Please find the authors' responses to each comment.

1. Since ocean heat content and thermometric anomalies are generally surface-intensified, so integrating from the surface to each depth of mapping does little to reveal how well the method works for mapping ocean heat content or thermometric anomalies at depths below the thermocline. In that respect it would be much more useful, and transparent, to map these quantities in distinct pressure or depth layers. Typical layers for using Argo might be 0-100 m, 100-300 m, 300-450 m, 450-700 m, 700-1000 m and 1000-2000 m, but others might be chosen if desiring more vertical resolution. The statistics for these would be much more revealing of the skill of the method below the surface layers than as it is currently formulated.

The primary objective of this work is to develop a model to estimate OHC accumulated between the sea surface to 700 m depth, which is the standard depth considered in reporting OHC (Domingues et al., 2008; Levitus et al., 2009, 2012; Balmaseda et al., 2013; Abraham et al., 2013; IPCC, 2014; Palmer et al., 2017; Cheng et al., 2017; Meyssignac et al., 2019; Irrgang et al., 2019). This depth covers most oceanic basins except shallower portions of bathymetry levels less than 700 m. As the OHC changes are more dynamic in shallower/upper oceanic layers, the current work has been extended to shallower depths lesser than 700 m and reported the models' performance at each depth extent in section 4.

On the other hand, OHC can be modeled based on the depth layers such as 0-20 m, 20 – 30 m, 30 – 40 m, …., and 650 – 700 m. It involves the layer-wise preparation of in-situ OHC databases, theoretical formulation derivation, collocation of satellite data of all the input parameters (based on the derived theoretical formulations), and development and validation of ANN models. Hence, we considered this layer-wise OHC modeling as a future scope of the current study.

2. The maps of ocean heat content (or thermometric sea level) are validated only against another machine learning estimate, but there are a number of objectively mapped and machine learning mapped products out there which are more widely used against which this work should be compared. See Johnson et al. (2023, https://doi.org/10.1175/BAMS-D-23-0076.2) and/or von Schuckman et al. (2023, https://doi.org/10.5194/essd-15-1675-2023) for an idea of the range of products available.

The main objective of the current study is to develop ANN models to provide daily OHC estimates from satellite data. Hence, direct measurements from the Argo systems are employed to compute in-situ OHC estimates rather than the interpolated/synthetic gridded data of monthly/quarterly/yearly scales to accurately capture physical oceanographic conditions during ANN training. To the best of our knowledge, OHC daily products are not available except the NRSC OHC products, and thus we compared our model results with the NNRSC OHC products (as presented in section 4.3). The mapped products derived from the ocean reanalysis systems and objectively analyzed techniques are of monthly to yearly temporal resolutions. Hence, we couldn't validate/compare our model results with other OHC products. However, the model-derived long-term mean distribution of OHC will be compared against monthly data of ARMOR3D and IAP products in section 5 (to be included in the revised manuscript).

3. Sea surface height is more directly related to both ocean heat content and of course thermosteric anomaly, so why aren't say CNEMS SSH maps, available since 1993, used as the neural network analysis?

Though SSH data have a profound relevance with OHC variations, critical issues associated with satellite altimetry-based SSH data in the precise implementation of the ocean thermal expansion method were identified and discussed in section 4.3. SSH (SLA) data must be corrected for haline and water mass changes (barotropic flows) before being used in OHC estimations (Sato et al., 2000; Jayne et al., 2003). Water mass change-related correction can be implemented in real-time from GRACE mission data on monthly scales, but not on daily scales. On the other hand, remote sensing-based approaches are still lacking in estimating the haline correction factors (halosteric sea level anomaly) in real-time. Moreover, the base period employed in preparing CMEMS SSH data and World Ocean Atlas-2018 climatological OHC differs from one to another (lines 57-73, 331-338).

The current study aims to provide a novel approach to implement the ocean thermal expansion method by eliminating all the potential sources of uncertainties as identified and reported in this study. Hence, the prime criterion followed in choosing the parameters for ANN modeling is based on the theoretical relationship between the input and output parameters rather than the direct usage of all the relevant parameters. The one-to-one relationship between OHC and TSL is employed in the OHC model. To arrive at TSL, the theoretical dependency of TSL on temperature and salinity is considered in TSL modeling work. This approach, when implemented in the current study, eliminates all the uncertainties associated with haline and water mass change corrections, base period, and other issues, as explained in sections 1 and 4.3.

4. Why are only Argo data used in the training? There are many more temperature profiles from shipboard CTDs, XBTs, marine mammals, and other sources readily available from the WOD that expand the spatial and temporal coverage Argo data considerably. Furthermore, Argo data should be downloaded directly from an Argo global data assembly centre, not WOD at NCEI, for the most recent version, as the data are subject to delayed-mode quality control and revisited frequently. As a result NCEI archives may be out-of-date.

The criteria considered in choosing the in-situ data sources are uniformity (in measuring techniques, instrumentation, and bias corrections), quality control (to eliminate erroneous or spurious data), and data coverage (in three-dimensional ocean space and time scales). Argo data have dense coverage in three-dimensional ocean space and time scales compared to other sources of CTD profile data. Employing Argo data alone suffice in achieving spatial and temporal coverage with uniformity.

Regarding the data used in this study, the Argo global data assembly centers distribute CTD data in separate files for each float at their observed depth levels. These depth levels vary from one float to another. WOD at NCEI provides the Argo-based objectively analyzed CTD profile data at standard depth levels in a single NetCDF file. Moreover, WOD provides quality flags for each measurement (Cheng and Zhu, 2016). All these considerations made the WOD data highly suited for reading, eliminating erroneous profiles, and preparing in-situ OHC estimates from Argo data (Levitus et al., 2012; Ali et al., 2012; Jagadeesh et al., 2015; Cheng et al., 2017).

5. The data section covers only the in situ data, and not the fields used for prediction. Please consider moving the description of the SST and SSS "data" into the data section. Also, hopefully SSH data too, as that should certainly be useful in constructing the network. Another point, the ORAS5 "SSS" data are not from satellite measurements! Those maps are output of an ocean reanalysis, that is to say, an assimilation of in situ data (including all the Argo profile data) into a numerical ocean model. So ORAS5 SSS fields are certainly not independent from Argo data, in fact, they directly incorporate those data. This fact, and the model used and datasets ingested, are all well documented on the ORAS5 websites. Please study the documentation and revise the analysis, manuscript, and claims within accordingly.

The description of satellite-derived data will be moved to "Data" section in the revised manuscript.

The source of ORAS5 SSS will be specified as Ocean reanalysis system in the revised manuscript when it refers to the ORAS5 SSS dataset.

Owing to the dependency of ORAS5 SSS data on Argo profiles, we have considered the different time periods during model training and validation. The SSS data used during model training is from the Argo measurements over the period 2005-2016 and satellite-based validation is from ocean reanalysis data over the period 2017-2020.

6. Section 4.1. Given the relatively large (100s to 1000s of km) and long (30-120 days) decorrelation scales of ocean temperature anomalies, it is misleading to call or treat these points as "independent". Almost all of them will be well correlated with some of the data used to construct the ANN. This probably means the true uncertainties are much higher than estimated here.

In the current study, the database has been divided into training (2005-2016) and testing/validation (2017-2020) based on the time period. This eliminates the correlation between the data used during model training and testing phases. However, the data used during late 2016 and early 2017 possess some correlation, which is in the order of a few weeks, and its effect on the reported validation metrics can be neglected. However, by considering this part of SST data and ORAS5 SSS data, the manuscript will be revised by referring testing/validation datasets as *unseen* datasets instead of independent datasets.

7. What is the point of section 5? It gives neither new oceanographic information nor insight into the accuracy or usefulness of the maps generated. It might better to compare and contrast trends with trends from other products. If a section on the topic is retained, it would be beneficial to recruit a co-author familiar with oceanography, as the section as written is haphazard.

Thank you for your critical points. We will revise section 5 by providing spatial distribution of long-term mean OHC estimates for the period 1993-2020. OHC estimates across all the standard depth levels considered from 20 m to 700 m can be merged based on the GEBCO-2020 grid to arrive at depth-varying OHC estimates. Further, the observed model-derived OHC estimates and annual time series will be compared with annual OHC estimates obtained from the ARMOR3D and IAP products. Among the available mapped CTD fields and OHC products, ARMOR3D and IAP products comprise temperature and salinity data at all standard depth levels that are considered in the current work except 150 m and 650 m depth levels.

Minor comments

8. L24. 93% is an old estimate at this point. See von Schuckman et al. (2023) for an update.
The percentage OHC absorbed by oceans will be updated as 89% in the revised manuscript.

9. L27. The statement that this understanding is "inevitable" seems an odd word choice. It is important, or societally relevant, but that does not make it inevitable.
The word 'inevitable' will be replaced with the phrase 'is of great importance to the scientific community' in the revised manuscript

10. L32-33. Actually, just temperature profiles with depth information are also useful (e.g. XBT data). Measurements of conductivity on all profiles are not strictly necessary.
Sea water density and conservative temperature (Meyssignac et al., 2019) employed in OHC computation are functions of in-situ temperature, salinity, and pressure (IOC et al., 2010). Hence, simultaneous measurements of in-situ-based temperature, salinity, and pressure (depth) profile data are employed in this study to estimate OHC changes from in-situ data.

11. L42-87. This paragraph is very long and rather stream-of-consciousness in style. Consider breaking it up into a few paragraphs covering the different topics to better organize the exposition.
This paragraph will be divided into smaller paragraphs.

12. L48. There is a recent Zhao manuscript on the subject that should be referenced.
Zhao (2017) manuscript titled "Propagation of the Semidiurnal Internal Tide: Phase Velocity Versus Group Velocity" will be referenced in the revised manuscript.

13. L177-178. Here and elsewhere in the manuscript, sentences like "blah blah blah are shown in Figure 3" generally duplicate figure captions and interrupt the flow of ideas. It's much better to write a sentence that communicates the main thing the authors want the readers to get out of the figure and refer to it parenthetically.
We will implement this modification throughout the revised manuscript.

14. L241-249. The more energetic regions of the oceans will of course be more difficult to model. It might be appropriate to reference studies that have documented the spatial variability of eddy energy in the oceans, highlighting the western boundary current extensions and Antarctic Circumpolar Current, among other regions.
We will provide references to Beech et al., 2022 and Ni et al., 2023 in the revised manuscript.

15. L466. The phrase "Future releases of WOA will certainly resolve the complex patterns..." again seems odd, arguably incorrect, and out of place. Please consider deleting or rephrasing this sentence.
This sentence will be removed in the revised manuscript as it is conveyed well enough in the previous statement "More number of in-situ CTD profiles need to be collected and analyzed in such oceanic regions to address the associated complex patterns".

---

## Author Response (AR1)

**Dear Editor,**

We are grateful for your immense efforts in contacting critical reviewers. We would like to extend our heartfelt thanks for the constructive reviewer and editorial comments.

We have revised the manuscript as per the review comments and suggestions. In addition, the scientific contents of the manuscript have been improved for better presentation and easy understanding to readers.

The following are the major changes incorporated in the revised manuscript:
1. Proposed OHC models were compared against the existing global OHC estimates such as NCEI, IAP, PMEL, and OPEN-LSTM products in section 5
2. Reproduced density scatterplots for better visualization (Figs. 4-7)

And, major justifications are provided for the following in the authors' responses and manuscript:
1. Depth-integrated modeling of OHC
2. Performance assessment
3. Avoiding SSH (SLA) in OHC modeling and prime criteria followed while choosing input parameters
4. In-situ CTD profile data from the Argo program and use of World Ocean Database-2018
5. Need for conductivity/salinity data in OHC computation

We are uploading the following files to the journal review submission webpage:
1. Point-by-point response to the reviewers' comments (below)
2. Clean version of the revised manuscript
3. Tracked version of the revised manuscript

We hope that the revised manuscript addresses all the comments of the reviewers and meets the journal's standards.

Sincerely,
Shanmugam

**Reviewer 1:**

Review of esd-2024-1, "Estimating ocean heat content from the ocean thermal expansion parameters using satellite data", submitted by Kondeti and Palanisamy for possible publication in Earth System Dynamics.

This manuscript presents a methodology for making maps of ocean heat content (and thermosteric sea level) using an artificial neural network that incorporates Argo CTD data together with what the authors identify as satellite SST and SSS data, as well as the results of the application of that method. There are some very major issues that will need to be resolved prior to publication. Specific minor comments follow, first major, then minor, indexed by line number (L) where possible.

Thank you for your constructive comments that have improved the manuscript. Please find the authors' responses to each comment.

1. Since ocean heat content and thermometric anomalies are generally surface-intensified, so integrating from the surface to each depth of mapping does little to reveal how well the method works for mapping ocean heat content or thermometric anomalies at depths below the thermocline. In that respect it would be much more useful, and transparent, to map these quantities in distinct pressure or depth layers. Typical layers for using Argo might be 0-100 m, 100-300 m, 300-450 m, 450-700 m, 700-1000 m and 1000-2000 m, but others might be chosen if desiring more vertical resolution. The statistics for these would be much more revealing of the skill of the method below the surface layers than as it is currently formulated.

The primary objective of this work is to develop a model to estimate OHC accumulated between the sea surface to 700 m depth, which is the common depth considered in reporting OHC (Domingues et al., 2008; Levitus et al., 2009, 2012; Balmaseda et al., 2013; Abraham et al., 2013; IPCC, 2014; Palmer et al., 2017; Cheng et al., 2017; Meyssignac et al., 2019; Irrgang et al., 2019). This depth covers the most oceanic basins except shallower portions of bathymetry levels less than 700 m. As the OHC changes are more dynamic in shallower/uppermost oceanic layers, the current work has been extended to shallower depths lesser than 700 m and reported the models' performance at each depth extent in section 4. Furthermore, 17 modeling depths considered in this study are chosen according to the OHC models developed for ice-covered oceanic regions (Prakash and Shanmugam, 2022). It enables the research community to generate satellite-based pan-global OHC maps by covering both open and ice-covered oceans of varying bathymetry levels ($\geq$ 20 m) (Lines 83-87 and 475).

On the other hand, OHC can be modeled based on the depth layers such as 0-20 m, 20 – 30 m, 30 – 40 m, ...., and 650 – 700 m (Lyman and Johnson, 2023). It involves the layer-wise preparation of in-situ OHC database, climatological OHC database, theoretical formulation derivation, collocation of satellite data of all the input parameters (based on the derived theoretical formulations), and development and validation of ANN models. Hence, we considered this layer-wise OHC modeling as a future scope of the current study.

2. The maps of ocean heat content (or thermometric sea level) are validated only against another machine learning estimate, but there are a number of objectively mapped and machine learning mapped products out there which are more widely used against which this work should be compared. See Johnson et al. (2023, https://doi.org/10.1175/BAMS-D-23-0076.2) and/or von Schuckman et al. (2023, https://doi.org/10.5194/essd-15-1675-2023) for an idea of the range of products available.

The main objective of the current study is to develop ANN models to produce daily OHC estimates from satellite data. Hence, direct (discrete) CTD observations from Argo floats are employed to compute in-situ OHC estimates rather than the mapped OHC products of daily/weekly/monthly/quarterly/yearly scales to accurately capture physical oceanographic conditions during ANN training and subsequent in-situ and satellite-based validations. To the best of our knowledge, contemporary OHC daily products are not available except the NRSC OHC products for 7 modeling depths, and thus, we assessed our model performance with the NRSC OHC products (as presented in section 4.3). The other OHC/TSL products derived from the objectively analyzed techniques and machine learning techniques are of weekly/monthly/quarterly/yearly temporal resolutions (Meyssignac et al., 2019; Lyman and Johnson, 2023; Von Schuckmann et al., 2023; Johnson et al., 2023; Su et al., 2020, 2021; Cheng et al., 2017; Balmaseda et al., 2013, 2015). Hence, we couldn't validate our model with other OHC/TSL products. However, the model-derived OHC spatial and time series distributions have been compared against annual data of NCEI (Levitus et al., 2012), IAP (Cheng et al., 2017), PMEL (Lyman and Johnson, 2023), and OPEN-LSTM (Su et al., 2021) products in section 5 of the revised manuscript.

3. Sea surface height is more directly related to both ocean heat content and of course thermosteric anomaly, so why aren't say CNEMS SSH maps, available since 1993, used as the neural network analysis?

Though SSH data have a profound relevance with OHC (TSL) variations, critical issues associated with CMEMS SSH data in the precise implementation of the ocean thermal expansion method were identified and discussed in sections 1 & 4.3 (Lines 57-73 and 334-344). SSH (SLA) data must be corrected for haline and water mass changes (barotropic flows) before being used in OHC estimations (Sato et al., 2000; Jayne et al., 2003). Water mass change-related correction can be implemented in real-time from GRACE mission data on monthly scales, but not on daily scales. On the other hand, remote sensing-based approaches are still lacking in estimating the haline correction factors (halosteric sea level anomaly) in real-time. And, the base period employed in preparing CMEMS SSH data and World Ocean Atlas climatological OHC differs from one to another. Hence, we didn't use CMEMS SSH data in OHC modeling to avoid the potential sources of uncertainties associated with haline and water mass change corrections and base periods. However, SLA and climatological OHC data of the same base period are desirable and can be used in OHC (TSL) modeling if available in the future.

The current study aims to provide a novel approach to implement the ocean thermal expansion method by eliminating all the potential sources of uncertainties as identified and reported in this study (Lines 57-73 and 334-344). Hence, the prime criterion followed in choosing the input parameters for ANN modeling is the theoretical relationship between the input and output parameters rather than the direct usage of all the relevant parameters. The one-to-one relationship between OHC and TSL is employed in the OHC modeling. To arrive at TSL, the theoretical

dependency of TSL on temperature and salinity is considered in TSL modeling work (Lines 150-155). This approach, when implemented in the current study, eliminates all the uncertainties associated with haline and water mass change corrections, base period, and other issues, as explained in sections 1 and 4.3 (Lines 57-73 and 334-344).

4. Why are only Argo data used in the training? There are many more temperature profiles from shipboard CTDs, XBTs, marine mammals, and other sources readily available from the WOD that expand the spatial and temporal coverage Argo data considerably. Furthermore, Argo data should be downloaded directly from an Argo global data assembly centre, not WOD at NCEI, for the most recent version, as the data are subject to delayed-mode quality control and revisited frequently. As a result NCEI archives may be out-of-date.

The criteria considered in choosing the in-situ data sources are uniformity (in measuring techniques, instrumentation, and bias corrections), quality control (to eliminate erroneous or spurious data), and data coverage (in three-dimensional ocean space and time scales). Argo data have dense coverage in three-dimensional ocean space and time scales compared to other sources of CTD profile data (Meyssignac et al., 2019). Employing Argo data alone suffice in achieving spatial and temporal coverage with uniformity.
      Regarding the data used in this study, the Argo global data assembly centers distribute CTD data in separate files for each float at their observed depth levels. Moreover, the observed depth levels vary from one float to another. WOD at NCEI provides the Argo CTD profile data of all the floats at standard depth levels in a single NetCDF file with quality flags for each measurement (Garcia et al., 2018, https://www.ncei.noaa.gov/sites/default/files/2020-04/wodreadme_0.pdf; Cheng et al., 2017). All these considerations made the WOD data highly suited for reading, eliminating erroneous profiles, and preparing in-situ OHC databases from Argo systems for all the 17 modeling depths considered (Levitus et al., 2012; Ali et al., 2012; Jagadeesh et al., 2015; Cheng et al., 2017; Cheng and Zhu, 2016).

5. The data section covers only the in situ data, and not the fields used for prediction. Please consider moving the description of the SST and SSS "data" into the data section. Also, hopefully SSH data too, as that should certainly be useful in constructing the network. Another point, the ORAS5 "SSS" data are not from satellite measurements! Those maps are output of an ocean reanalysis, that is to say, an assimilation of in situ data (including all the Argo profile data) into a numerical ocean model. So ORAS5 SSS fields are certainly not independent from Argo data, in fact, they directly incorporate those data. This fact, and the model used and datasets ingested, are all well documented on the ORAS5 websites. Please study the documentation and revise the analysis, manuscript, and claims within accordingly.

The description of satellite-derived data has been moved to "Data" section in the revised manuscript (Lines 128-139).
      The source type of ORAS5 SSS is specified as "Ocean reanalysis system" in the revised manuscript when it refers to the ORAS5 SSS dataset (Lines 131, 135, 275, 321, and 481).
      Owing to the dependency of ORAS5 SSS data on Argo profiles, we have considered the different time periods during model training and validation. The SSS data used for model training come from direct Argo measurements over the period 2005-2016; the in-situ-based validation dataset comes from direct Argo measurements over the period 2017-2020, and the satellite-based validation is based on ORAS5 data over the period 2017-2020 (Lines 120-124 and 128-133). These

datasets enabled better evaluation of the model performance on unseen data in both in-situ and satellite-based validations.

6. Section 4.1. Given the relatively large (100s to 1000s of km) and long (30-120 days) decorrelation scales of ocean temperature anomalies, it is misleading to call or treat these points as "independent". Almost all of them will be well correlated with some of the data used to construct the ANN. This probably means the true uncertainties are much higher than estimated here.

In the current study, the database has been divided into training (2005-2016) and testing/validation (2017-2020) based on the time period (Momin et al., 2011; Jagadeesh et al., 2015). This eliminates the correlation between the data used during model training and validation phases. However, the data used during late 2016 and early 2017 possess some correlation, which is in the order of a few weeks, and its effect on the reported validation metrics can be neglected. The same can be observed from the spatial maps showing percentage error distribution (Figs. A2 and A3). However, by considering this part of SST data and ORAS5 SSS data, the manuscript has been revised by referring the testing/validation datasets as *unseen* datasets instead of independent datasets (Lines 225, 231, 264, 269, 300, 325, 373, and 475).

7. What is the point of section 5? It gives neither new oceanographic information nor insight into the accuracy or usefulness of the maps generated. It might better to compare and contrast trends with trends from other products. If a section on the topic is retained, it would be beneficial to recruit a co-author familiar with oceanography, as the section as written is haphazard.

Thank you for your critical points. We have revised section 5 by presenting long-term trends and time series distribution of model-derived OHC estimates for the period 1993-2020. The OHC estimates across all the modeling depth levels considered from 20 m to 700 m have been merged based on the GEBCO-2020 grid to arrive at depth-varying OHC estimates. Further, the observed model-derived OHC trends and annual time series were compared with annual OHC estimates obtained from the NCEI (Levitus et al., 2012), IAP (Cheng et al., 2017), PMEL (Lyman and Johnson, 2023), and OPEN-LSTM (Su et al., 2021) products.

**Minor comments**

8. L24. 93% is an old estimate at this point. See von Schuckman et al. (2023) for an update.
The percentage OHC absorbed by oceans is updated as 89% in the revised manuscript (Line 24).

9. L27. The statement that this understanding is "inevitable" seems an odd word choice. It is important, or societally relevant, but that does not make it inevitable.
The word 'inevitable' is replaced with the phrase 'is of great importance to the scientific community' in the revised manuscript (Lines 27-28).

10. L32-33. Actually, just temperature profiles with depth information are also useful (e.g. XBT data). Measurements of conductivity on all profiles are not strictly necessary.
Sea water density and conservative temperature (Meyssignac et al., 2019) employed in OHC computation are functions of in-situ temperature, salinity, and pressure (IOC et al., 2010) (Lines 111-118). Hence, simultaneous measurements of in-situ-based temperature, salinity, and pressure

(depth) profile data are preferable and the same were employed in the current study to estimate OHC changes from in-situ data.

11. L42-87. This paragraph is very long and rather stream-of-consciousness in style. Consider breaking it up into a few paragraphs covering the different topics to better organize the exposition. This paragraph has been divided into smaller paragraphs (Lines 43-91).

12. L48. There is a recent Zhao manuscript on the subject that should be referenced. Zhao (2017) manuscript titled "Propagation of the Semidiurnal Internal Tide: Phase Velocity Versus Group Velocity" is referenced in the revised manuscript (Line 49).

13. L177-178. Here and elsewhere in the manuscript, sentences like "blah blah blah are shown in Figure 3" generally duplicate figure captions and interrupt the flow of ideas. It's much better to write a sentence that communicates the main thing the authors want the readers to get out of the figure and refer to it parenthetically. We have implemented this modification throughout the revised manuscript (Lines 196, 237-239, 249, etc wherever applicable).

14. L241-249. The more energetic regions of the oceans will of course be more difficult to model. It might be appropriate to reference studies that have documented the spatial variability of eddy energy in the oceans, highlighting the western boundary current extensions and Antarctic Circumpolar Current, among other regions. We have provided the references to Beech et al., (2022) and Ni et al., (2023) in the revised manuscript (Line 252).

15. L466. The phrase "Future releases of WOA will certainly resolve the complex patterns..." again seems odd, arguably incorrect, and out of place. Please consider deleting or rephrasing this sentence. This sentence is removed in the revised manuscript as the same point is conveyed well enough in the previous statement "More number of in-situ CTD profiles need to be collected and analyzed in such oceanic regions to address the associated complex patterns" (Lines 491-492).

**Reviewer 2:**

This manuscript has the potential to provide some key strengths, by seeking to contribute to addressing the thorny problem of geoinformation retrieval from satellite data through information-technologically assisted inverse modeling: more precisely ocean "heat" content from proxy parameters derived from satellite data and assisted by artificial neural network data-based modeling.

Overall, the initiative is welcome and courageous, given the minefield of caveats across the retrieval and estimation workflow, which can easily hinder the accuracy and reliability of the formal results and condition the confidence with which novel insights can effectively be drawn from the whole process.

Notwithstanding the complexity and fragility of the overall procedure, the authors have made efforts to document the workflow and underlying reasoning. That does not mean that the results can be taken for factual and fail-proof. However, that does not mean that they should be entirely discarded either without a further attempt to strengthen the methodological and operational robustness of the study workflow, namely in terms of enabling assumptions, limitations and implications.

(For instance, Pearson correlation suffers from various disqualifying shortcomings, especially when relating such complex data, that the reliance on such basic linear metric for relating environmentally relevant datasets is highly questionable).

To that end, it will be crucial to aptly and thoroughly explain and clarify all the fundamental and operational assumptions, limitations and respective mitigation strategies. That is required to more confidently be able to legitimize the research procedure and allow for interpretable results to be drawn and effectively assessed as valid outcomes. While the authors have invested considerable efforts in explaining such in the original manuscript, further diligence is needed, as also pointed out by the other reviewer.

Crucial attention needs to be placed in the formal algorithmic procedures, the underlying mathematical details, and the physical justifications for each conducted step, so that a more detailed feedback can be provided pertaining the associated intricacies and underpinnings beyond these more general remarks. Currently, as also shared by the other referee, major concerns hover over the solidity of the research procedure, which will require thorough clarification and revision.

On a more minor (yet still relevant) aspect, the density scatterplots require some visualization improvement, namely in terms of color scale, so that the different shades can be more easily seen and grasped.

While not repeating the concerns of the other reviewer, I reiterate my overall agreement with them and the need to overcome such for strengthening the contribution. Notwithstanding the difficulties faced by the submitted study and the lack of detail that precludes a clear conclusion on the validity of the procedure and quality of the data (which require thorough clarification and exploitation in

the revised version), I hope that the authors can clarify the questions, overcome the shortcomings and produce a much more solid revised manuscript.

Thank you for your critical points about strengthening the manuscript. The motivation for the current study is to address the challenges associated with satellite-based estimation of OHC. To achieve this, we have conducted a thorough literature to understand the deficiency in OHC estimations and to identify the shortcomings in the existing works. It is observed that the OHC modeling needs a solution that involves accurate computation of in-situ OHC from the CTD profiles, precise theoretical formulations for choosing the remote sensing-based parameters and their accurate retrievals, the latest version of climatological data, efficient modeling tools, and appropriate validation methods. To the best of our knowledge, we have made a successful attempt by fulfilling all the considerations and explained the same in the manuscript. However, we have revised the manuscript thoroughly by better presenting the theoretical formulations (mathematics and physics involved), methodology, data, modeling techniques, validation methods, figures (density scatterplots, Figs. 4-7), limitations, and future directions.

**References cited in authors' responses:**

Abraham, J. P., Baringer, M., Bindoff, N. L., Boyer, T., Cheng, L. J., Church, J. A., Conroy, J. L., Domingues, C. M., Fasullo, J. T., Gilson, J., Goni, G., Good, S. A., Gorman, J. M., Gouretski, V., Ishii, M., Johnson, G. C., Kizu, S., Lyman, J. M., Macdonald, A. M., Minkowycz, W. J., Moffitt, S. E., Palmer, M. D., Piola, A. R., Reseghetti, F., Schuckmann, K., Trenberth, K. E., Velicogna, I., and Willis, J. K.: A review of global ocean temperature observations: Implications for ocean heat content estimates and climate change, Rev. Geophys., 51, 450–483, https://doi.org/10.1002/rog.20022, 2013.

Ali, M. M., Jagadeesh, P. S. V., Lin, I. I., and Hsu, J. Y.: A neural network approach to estimate tropical cyclone heat potential in the Indian Ocean, IEEE Geosci. Remote Sens. Lett., 9, 1114–1117, https://doi.org/10.1109/LGRS.2012.2190491, 2012.

Balmaseda, M. A., Trenberth, K. E., and Källén, E.: Distinctive climate signals in reanalysis of global ocean heat content, Geophys. Res. Lett., 40, 1754–1759, https://doi.org/10.1002/grl.50382, 2013.

Balmaseda, M. A., Hernandez, F., Storto, A., Palmer, M. D., Alves, O., Shi, L., Smith, G. C., Toyoda, T., Valdivieso, M., Barnier, B., Behringer, D., Boyer, T., Chang, Y. S., Chepurin, G. A., Ferry, N., Forget, G., Fujii, Y., Good, S., Guinehut, S., Haines, K., Ishikawa, Y., Keeley, S., Köhl, A., Lee, T., Martin, M. J., Masina, S., Masuda, S., Meyssignac, B., Mogensen, K., Parent, L., Peterson, K. A., Tang, Y. M., Yin, Y., Vernieres, G., Wang, X., Waters, J., Wedd, R., Wang, O., Xue, Y., Chevallier, M., Lemieux, J. F., Dupont, F., Kuragano, T., Kamachi, M., Awaji, T., Caltabiano, A., Wilmer-Becker, K., and Gaillard, F.: The ocean reanalyses intercomparison project (ORA-IP), J. Oper. Oceanogr., 8, s80–s97, https://doi.org/10.1080/1755876X.2015.1022329, 2015.

Beech, N., Rackow, T., Semmler, T., Danilov, S., Wang, Q., and Jung, T.: Long-term evolution of ocean eddy activity in a warming world, Nat. Clim. Chang., 12, 910–917, https://doi.org/10.1038/s41558-022-01478-3, 2022.

Cheng, L. and Zhu, J.: Benefits of CMIP5 multimodel ensemble in reconstructing historical ocean subsurface temperature variations, J. Clim., 29, 5393–5416, https://doi.org/10.1175/JCLI-D-15-0730.1, 2016.

Cheng, L., Trenberth, K. E., Fasullo, J., Boyer, T., Abraham, J., and Zhu, J.: Improved estimates of ocean heat content from 1960 to 2015, Sci. Adv., 3, 1–10, https://doi.org/10.1126/sciadv.1601545, 2017.

Domingues, C. M., Church, J. A., White, N. J., Gleckler, P. J., Wijffels, S. E., Barker, P. M., and Dunn, J. R.: Improved estimates of upper-ocean warming and multi-decadal sea-level rise, Nature, 453, 1090–1093, https://doi.org/10.1038/nature07080, 2008.

IOC, SCOR, and IAPSO: The international thermodynamic equation of seawater-2010: Calculation and use of thermodynamic properties Intergovernmental Oceanographic Commission, 2010.

IPCC: Climate Change 2014: Synthesis Report. Contribution of Working Groups I, II and III to the Fifth Assessment Report of the Intergovernmental Panel on Climate Change, J. Cryst. Growth, 2014.

Irrgang, C., Saynisch, J., and Thomas, M.: Estimating global ocean heat content from tidal

magnetic satellite observations, Sci. Rep., 9, 1–8, https://doi.org/10.1038/s41598-019-44397-8, 2019.

Jagadeesh, P. S. V., Suresh Kumar, M., and Ali, M. M.: Estimation of Heat Content and Mean Temperature of Different Ocean Layers, IEEE J. Sel. Top. Appl. Earth Obs. Remote Sens., 8, 1251–1255, https://doi.org/10.1109/JSTARS.2015.2403877, 2015.

Jayne, S. R., Wahr, J. M., and Bryan, F. O.: Observing ocean heat content using satellite gravity and altimetry, J. Geophys. Res. Ocean., 108, 1–12, https://doi.org/10.1029/2002jc001619, 2003.

Johnson, G. C. and R. Lumpkin, E.: Global Oceans [in "State of the Climate in 2022"], in: Bulletin of the American Meteorological Society, vol. 104, S146–S206, https://doi.org/https://doi.org/10.1175/BAMS-D-23-0076.2, 2023.

[revised manuscript text omitted]